# VULNERABLE REGION DISCOVERY THROUGH DIVERSE ADVERSARIAL EXAMPLES

## ABSTRACT

Deep Neural Networks (DNNs) have shown great promise in multiple fields, but ensuring their reliability remains a challenge. Current explainable approaches for DNNs mainly aim at understanding DNNs' behavior by identifying and prioritizing the influential input features that contribute to the model's predictions, often overlooking *vulnerable regions* that are highly sensitive to small perturbations. Traditional norm-based adversarial example generation algorithms, due to their lack of spatial constraints, often distribute adversarial perturbations throughout images, making it hard to identify these specific vulnerable regions. To address this oversight, we introduce an innovative method that uncovers these vulnerable regions by employing adversarial perturbations at diverse locations. Specifically, our method operates within a one-pixel paradigm. This enables detailed pixel-level vulnerability assessments by evaluating the effects of individual perturbations on predictions. By leveraging the robust Sharing Differential Evolution Algorithm, we can simultaneously identify multiple one-pixel perturbations, forming a vulnerable region. We conduct thorough experiments across a variety of network architectures and adversarial training techniques, showing that our approach not only effectively identifies vulnerable regions but also provides invaluable insights into the inherent vulnerabilities present in a diverse range of deep learning models.

## 1 INTRODUCTION

Deep Neural Networks (DNNs) have led to groundbreaking advancements in various complex fields, such as computer vision and natural language processing, setting new benchmarks for both accuracy and efficiency He et al. (2016); Pouyanfar et al. (2017); Vaswani et al. (2017); Liu et al. (2021). Despite their remarkable proficiency, the challenge of achieving a comprehensive and reliable understanding of these networks remains a pressing issue. Much of the existing research on explainable DNNs has focused on identifying the salient features that influence DNNs' decisions Zeiler & Fergus (2014); Zhou et al. (2016); Selvaraju et al. (2017); Yang et al. (2021).

While significant strides have been made, a notable challenge remains: pinpointing specific network regions particularly vulnerable to subtle adversarial manipulations. A straightforward approach for identifying these vulnerable regions is by analyzing the locations where perturbations effectively deceive the DNNs. Common norm-based methods, namely, $\ell_0, \ell_2$, and $\ell_\infty$ approaches for creating adversarial examples, come with inherent constraints. Those that allow unrestricted perturbations, such as the $\ell_2$ and $\ell_\infty$ methods, often distribute perturbations densely across images, obscuring the identification of specific vulnerable regions Carlini & Wagner (2017a); Moosavi-Dezfooli et al. (2017); Madry et al. (2018). On the other hand, methods that limit the number of pixel changes (i.e., $\ell_0$ methods) Modas et al. (2019); Croce et al. (2022) tend to distribute perturbations sparsely across images. These methods rely on the collective effect of the diffused perturbations to deceive DNNs, complicating the task of analyzing the true vulnerabilities and their severity in specific vulnerable regions. Specifically, the one-pixel attack Su et al. (2019) focuses only on the most harmful pixel, limiting its ability to fully reveal these critical vulnerable regions.

Our work addresses the limits of current large-scale and sparse perturbation methods by utilizing diversely located one-pixel adversarial perturbations to occupy vulnerable regions. The characteristic of this one-pixel approach simplifies attributing DNN errors to distinct input points. This lets us assess vulnerability based on the influence of these one-pixel perturbations on DNN predictions, as illustrated in Fig. 1. Additionally, varying the perturbation locations ensures a comprehensive reve-

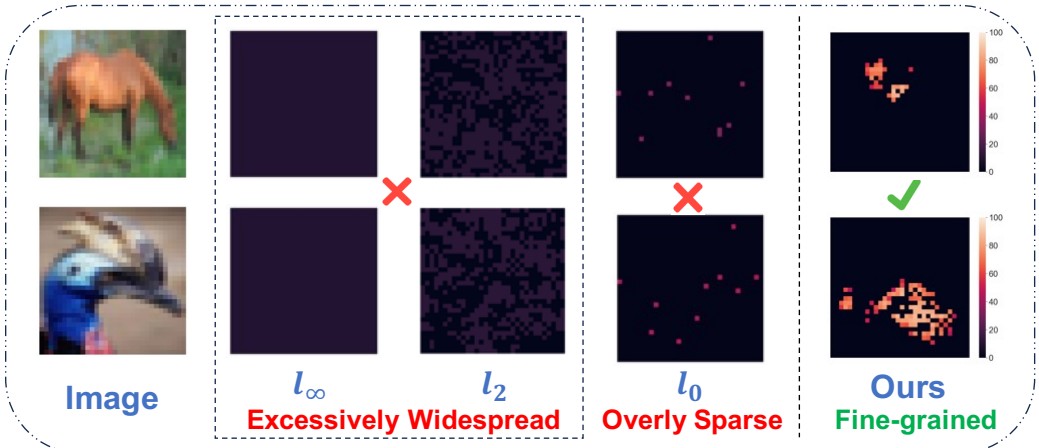

Figure 1: Comparing with three different perturbations generated by PGD ($\ell_\infty$) Madry et al. (2018), CW ($\ell_2$) Carlini & Wagner (2017b), and RSAttack($\ell_0$) Croce et al. (2022) for vulnerable region discovery. Since these methods incorporate different perturbed pixels, we evenly distribute the reduced confidence of the ground truth across perturbed pixels. Notably, as perturbations generated by the PGD attack encompass the entire image, each pixel is thought to contribute just a little. In contrast, our approach reveals detailed vulnerable regions with a bar that indicates the level of vulnerability.

lation of potential vulnerabilities, offering a thorough analysis of these susceptible areas. To avoid the expensive computational cost of a brute-force approach that enumerates each pixel value, we leverage a powerful evolutionary algorithm-Sharing Differential Evolution. This algorithm is particularly efficient, evolving a set of candidate solutions toward a specific optimization direction and obtaining multiple solutions simultaneously. Furthermore, its unique sharing mechanism reduces the likelihood of generating similar solutions, such as pixels located nearby, ensuring the production of diverse one-pixel perturbations. The contribution can be summarized as follows:

- We propose a novel approach to identify vulnerable regions in DNNs through the use of diversely located adversarial perturbations. Specifically, we focus on a one-pixel scenario, offering a pixel-level vulnerability assessment.

- We adapt a powerful evolutionary algorithm to efficiently generate a set of diverse adversarial examples, without requiring access to the internal information of DNNs.

- Our extensive experiments demonstrate the effectiveness of our approach, yielding not only diverse adversarial examples but also providing novel insights into the inherent vulnerabilities of DNNs.

## 2 RELATED WORK

### 2.1 PERTURBATION-BASED METHODS FOR UNDERSTANDING DNNS

Perturbation-based methods have emerged as powerful tools to delve deeper into the intricacies of Deep Neural Networks (DNNs), mainly by altering the input and observing the resulting shifts in predictions. The Occlusion method Zeiler & Fergus (2014); Petsiuk et al. (2018) employs a gray square mask on parts of the input image and examine the variations in model predictions. LIME Ribeiro et al. (2016), leverages the occlusion of superpixels and uses linear models to accurately emulate the decision boundaries defined by the original deep model. However, the obtained saliency maps often lack precision due to the coarse-grained nature of superpixels. Taking the morphology of the objects into consideration, Fong et al. (2019) and Yang et al. (2021) both introduce optimization methods to iteratively refine the obtained saliency map. Despite the proficiency of these methods in identifying critical regions in an input image relative to the DNN output, they exhibit limitations in exposing any vulnerabilities of DNNs. As such, our research shifts focus towards the identification of these latent vulnerable regions within DNNs.

### 2.2 ADVERSARIAL VULNERABILITIES

Adversarial attacks, which attempt to fool deep models by exploiting their adversarial vulnerability, can typically be categorized into white-box and black-box attacks. A white-box attack means that

attackers have full access to the DNN Madry et al. (2018); Carlini & Wagner (2017b). Black-box attacks assume that attackers are limited to only the DNN input and output values, making it necessary to query the target DNN model (as a black box) a large number of times Chen et al. (2017); Ilyas et al. (2018); Guo et al. (2019). Recent studies on black-box attacks have prioritized improving the efficiency of adversarial example generation Andriushchenko et al. (2020); Sun et al. (2022); Shi et al. (2022); Bai et al. (2023). While such research excels at finding a single adversarial example to attack the DNN, it does not reveal any other vulnerability of DNNs. It is challenging to pinpoint the specific locations and characteristics of potential DNN vulnerabilities.

Few-pixel attacks Rao et al. (2020); Croce et al. (2022), which aim to deceive DNNs using the fewest perturbed pixels, highlighting that certain pixels play a pivotal role in misleading DNNs. Nonetheless, pinpointing the exact influence of these perturbed pixels on inducing incorrect predictions remains a significant challenge. An extreme form of this, the one-pixel attack, was introduced by Su et al. (2019). This study primarily demonstrates the feasibility of such an extreme attack without providing detailed analyses of the regions susceptible to these minor perturbations. To bridge this gap, we employ the Sharing Differential Evolution method to craft diverse adversarial one-pixel perturbations at varied locations, facilitating a more comprehensive analysis of the potential vulnerable areas within the networks.

## 3 PROPOSED APPROACH

### 3.1 PROBLEM DEFINITION

Consider a data pair $(x, y)$ and a classifier $C(x) = \arg \max_i f(x)_i$, where $f(x)_i$ is the confidence score of the i-th class predicted by the DNN. If the DNN makes a correct prediction, we have $C(x) = y$. The goal of the adversary is to generate the adversarial example $x' = x + \delta$ and deceive the classifier to give a false prediction $C(x') \neq y$. The perturbations $\delta$ are normally bounded by the $\ell_p$ norm for visual invisibility. Then the search problem can be transformed into an optimization problem with constraints Szegedy et al. (2013):

$$\arg \min_{\delta} \quad f(x + \delta)_y \qquad s.t. \quad ||\delta||_p \leq \epsilon_p, \tag{1}$$

where $y$ is the ground truth label and $\epsilon_p$ is the maximum allowed perturbation strength. In our approach, we allow perturbation of only one pixel, aiming to obtain a more precise assessment of vulnerability levels. Unlike previous work, which generates a single adversarial example for each test image, our approach identifies a set of diverse adversarial perturbations, denoted as $\mathbb{A} = \{\delta_1, \delta_2, ..., \delta_n\}$, to ensure that:

$$d(\delta_i, \delta_j) \neq 0 \qquad s.t. \quad \delta_i, \delta_j \in \mathbb{A} \tag{2}$$

Here, $d(\cdot)$ represents the spatial distance (Euclidean distance) between two perturbed one-pixels.

### 3.2 GENERATION OF DIVERSE ADVERSARIAL EXAMPLES

Differential Evolution (DE) Storn & Price (1997), a population-based algorithm, has emerged as a potent tool for tackling complex optimization challenges. It has demonstrated success in addressing a wide range of scientific problems, particularly in the realm of black-box problems. In this framework, we represent the population of candidate solutions as $\mathbb{S} \in \mathbb{R}^{P \times D}$, where $P$ represents the population size, and $D$ corresponds to the dimensionality of the optimization problem. Throughout the evolution of the algorithm, these solutions are iteratively refined using mutation, recombination, and selection for optimization.

In the realm of one-pixel attacks, each candidate solution, represented as $z_i \in \mathbb{S}$, can be encoded as a 5-element tuple encapsulating the coordinates alongside the RGB values of the perturbed pixel in question. During the mutation process, a new perturbed pixel, termed the mutant vector $v_i \in \mathbb{V}$, is generated for each perturbed pixel $z_i \in \mathbb{S}$ based on a weighted differential scheme:

$$v_i = z_{r_1} + W(z_{r_2} - z_{r_3}) \qquad s.t. \quad r_1 \neq r_2 \neq r_3 \tag{3}$$

where $W$ is a positive scaling factor to control the scale of the difference vector. The indices $r_1, r_2, r_3$ are mutually exclusive random integers within the range $[1, P]$. Then a trial vector $u_i \in \mathbb{U}$

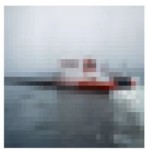 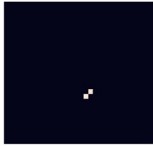 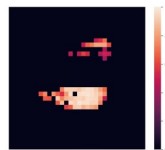 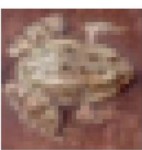 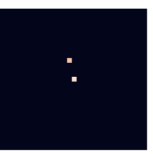 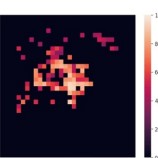

Figure 2: The final solutions obtained using one-pixel attack Su et al. (2019) and our algorithm 'on $32 \times 32$ images of CIFAR-10. **Three columns for each example correspond to the original image, the result of 10 independent one-pixel attacks, and the result of a single run of our algorithm.** Each point on the heatmap represents a successful one-pixel perturbation, with the brightness of the color indicating the reduction of confidence for the ground truth.

---

**Algorithm 1:** Generation of Diverse One-pixel Perturbations

---

**Input:** Selected image $I$, deep neural network $f$ for querying, population size $P$, number of generations $N$, mutation rate $W$, crossover rate $R$, sharing radius $\gamma$, hyperparameter $\alpha$

1: Initialize the candidate solutions for the first generation, denoted as $\mathbb{S}_0 \in \mathbb{R}^{P \times 5}$.
2: Evaluate raw fitness $F(\mathbb{S}_0)$ by querying $f$ with Image $I$ and perturbed pixels $\boldsymbol{z} \in \mathbb{S}_0$.
3: **for** $g = 1, ..., N$ **do**
4:     Execute the mutation process to obtain the set of mutant vectors $\mathbb{V}$ using Eq. 3.
5:     Execute the recombination process to obtain the set of trial vectors $\mathbb{U}$ using Eq. 4.
6:     Evaluate raw fitness $F(\mathbb{U})$ by querying $f$ with Image $I$ and perturbed pixels $\boldsymbol{u} \in \mathbb{U}$.
7:     Merge $\mathbb{U}$ and $\mathbb{S}_{g-1}$ to create the extended set $\mathbb{S}'_{g-1} \in \mathbb{R}^{2P \times 5}$.
8:     Compute the Euclidean distance between each pair of perturbed pixels in $\mathbb{S}'_{g-1}$.
9:     Calculate the sharing fitness $F'(\mathbb{S}'_{g-1})$ using Eqs. 5 and 6.
10:    Discard the worst half of the population based on sharing fitness $F'(\mathbb{S}'_{g-1})$ to form the new generation $\mathbb{S}_g$.
11:    **if** The individual with the highest raw fitness in $\mathbb{S}'_{g-1}$ is removed **then**
12:        Replace the least fit individual in $\mathbb{S}_g$ with this removed individual.
13:    **end if**
14:    Record the raw fitness $F(\mathbb{S}_g)$.
15: **end for**
**Output:** Return the population of the final generation $\mathbb{S}_N$.

---

is generated via the recombination of the mutant vector $\boldsymbol{v}_i$ and its corresponding parent vector $\boldsymbol{z}_i$:

$$\boldsymbol{u}_{i,j} = F \begin{cases} \boldsymbol{v}_{i,j}, & c \leq R \text{ or } j = j' \\ \boldsymbol{z}_{i,j}, & \text{otherwise} \end{cases} \tag{4}$$

where $c$ is a random number within the range [0,1]. The subscript $(i, j)$ denotes the $j$-th variable of the $i$-th individual. And $R$ is a hyperparameter known as the crossover rate, indicating the likelihood that a variable in the trial vector originates from the corresponding variable in the mutant vector. $j'$ is an integer randomly sampled from $[1, D]$ and is used to make sure at least one of the variables of the trial vector comes from the mutant vector. Subsequently, each trial vector is employed to substitute the corresponding pixel in the original image, facilitating an assessment of the extent to which it impairs the DNN's ability to accurately recognize the selected image. The generated solution $u_i$ will replace its corresponding parent solution $z_i$ if it leads to greater impairment in the DNN's recognition. However, such algorithms are designed to converge towards a particular optimal solution. As illustrated in Fig. 2, only a very small number of pixels have been pinpointed as viable for such a one-pixel attack Su et al. (2019), thereby making them unsuitable for uncovering vulnerable regions. To address this issue, we adapt a sharing differential evolution for the diversely located one-pixel perturbations.

The sharing mechanism for the evolution algorithm (EA) was introduced in Goldberg & Richardson (1987) with the goal of locating diverse solutions simultaneously. The core idea of the fitness sharing is to penalize individuals for occupying the same regions by applying a cost to their fitness scores. In general, fitness sharing serves to reduce the payoff of the densely populated area by dividing the raw fitness of an individual by the approximate number of similar individuals Darwen & Yao (1996). The shared fitness $F'_i$ of individual i is typically calculated as shown below:

$$F'_i = \frac{F_i}{\sum_{j=1}^{P} sh(d_{i,j})} \tag{5}$$

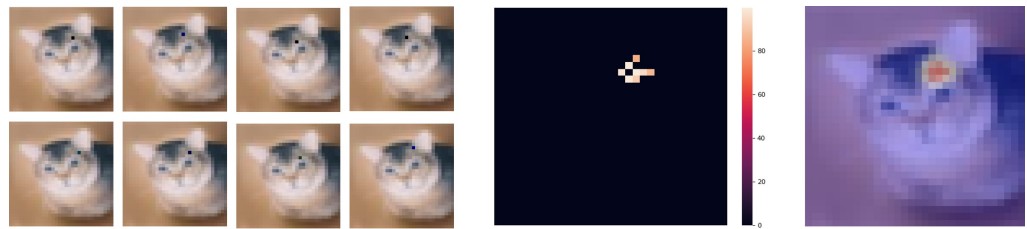

Figure 3: Diverse adversarial examples and the corresponding vulnerable region. The set of 8 small images shows the generated adversarial examples. The middle heatmap indicates one-pixel perturbation locations, with colors denoting reduced confidence in the 'cat' label. The rightmost image presents the heatmap overlaid on the original image, enhanced with Gaussian smoothing.

where $F_i$ is the original fitness score and p is the number of individuals included in the population. In the context of diverse one-pixel attack strategies, we define the fitness function $F_i$ as $1 - f(I')_y$ for non-targeted attacks, and $F_I = f(i')_t$ for targeted attacks. Here, $f(I')_y$ and $f(I')_t$ stand for the predict confidence in the ground truth label and a designated false class, respectively. We use $I'$ to denote the image disturbed by a one-pixel perturbation from either $\mathbb{S}$ or $\mathbb{U}$, and employ $sh(\cdot)$ as the sharing function defined as follows::

$$sh(d_{i,j}) = \begin{cases} 1 - (d_{i,j}/\gamma_s)^\alpha, & d < \gamma_s \\ 0, & \text{otherwise} \end{cases} \qquad (6)$$

where $d_{i,j}$ denotes the distance between the $i$-th individual and the $j$-th individual, $\gamma_s$ is the sharing radius and $\alpha$ controls the shape of the sharing function. In our work, we utilize Euclidean distance to prevent the algorithm from converging to pixels in the same position for obtaining diverse solutions.

In addition to the fitness function, the sharing DE algorithm modifies conventional DE in the following way. Instead of replacing the corresponding parents, all the newly generated offspring are added to the population to obtain sharing fitness value $F'$. Subsequently, we remove the worst half to stabilize the population size for the next generation. The Elitism ensures the preservation of the best-found solution throughout the optimization process: the best solution with original fitness $F$ are used to replace the worst individual in the population if it gets removed with scaled fitness $F'$. The searching process is presented in Algorithm 1.We denote raw fitness as $F(\cdot)$ and sharing fitness as $F'(\cdot)$. And they are specifically applicable to the set elements indicated within the brackets, providing the object value for the individual selection process.

## 4 EXPERIMENTAL STUDIES

We conduct extensive experiments using PyTorch on a single Tesla V100 GPU with two public image datasets: CIFAR-10 and ImageNet. For CIFAR-10, we evaluate our algorithm on VGG16 (86.04% accuracy), ResNet18 (94.03% accuracy), and Network in Network (91.49% accuracy). For ImageNet, we test on AlexNet (56.5% accuracy) and ResNet50 (75.9% accuracy). The results shown are averages from three independent experiments, involving non-targeted attacks to pinpoint vulnerable regions regardless of false predictions, and targeted attacks to uncover specific original-target class vulnerabilities. Throughout the following discussion, we refer to 'images with vulnerable regions' as 'vulnerable images'. Implemented details can be found in Appendix A.2.

### 4.1 DISCOVERING VULNERABLE REGIONS WITH DIVERSE ADVERSARIAL EXAMPLES

#### 4.1.1 DIVERSE ADVERSARIAL EXAMPLES WITH CIFAR-10

In the non-targeted and targeted attack scenarios, we randomly sample 1,000 and 500 correctly classified images from the CIFAR-10 test set, respectively. Specifically, for the targeted attack scenario, we attempt to perturb each of the selected 500 images into each of the other 9 categories. Results are presented in Table 1. For visualization of the vulnerable regions, the heatmap is overlaid on top of the images. The value of the heatmap represents the change in confidence of the class label. We also apply a Gaussian filter to the heatmap for better visualization. Examples of discovered vulnerable regions are shown in Fig. 4. Additional examples can be found in Fig 12 of Appendix A.7. Furthermore, to highlight the superiority of our algorithm in pinpointing diverse one-pixel perturba-

Table 1: Overall results on CIFAR-10 dataset. Left: Non-targeted attacks Right: Targeted attacks

| Non-targeted | ResNet18 | NiN | VGG16 | Targeted | ResNet18 | NiN | VGG16 |
|---|---|---|---|---|---|---|---|
| Success rate | 28.4% | 31.6% | 60.2% | Success rate | 5.0% | 5.6% | 13.2% |
| Pixel number | 37.3 | 40.8 | 41.7 | Pixel number | 29.4 | 27.9 | 20.6 |

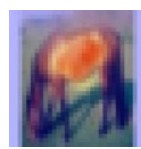 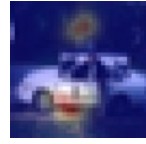 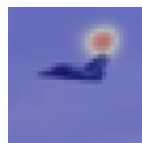 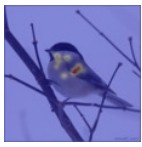 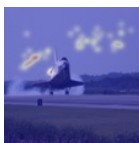 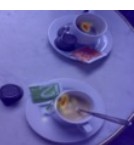

Figure 4: Heatmaps of vulnerable regions on CIFAR-10 (left) and ImageNet (right)

tions, we compare it with ten runs of the one-pixel attack Su et al. (2019). Results in Appendix A.3 demonstrates the effectiveness of our approach in identifying vulnerable regions.

**Diverse adversarial examples with non-targeted scenario**   Among the three DNN types we evaluated, our algorithm consistently demonstrated proficiency in generating diverse one-pixel adversarial examples. Notably, VGG16 emerged as the most vulnerable, with a success rate of 60.2%, almost twice as high as the 31.6% rate of the NiN network. This high success rate suggests that a significant portion of data points lies near the decision boundary along a single dimension. While success rates vary among the three DNNs, indicating the distinct likelihood of discovering vulnerable regions, the number of detected one-pixel perturbations for each vulnerable image exhibits remarkable similarity, with a maximum difference of 4.4 between VGG16 and ResNet18.

**Diverse adversarial examples with targeted scenario**   With the lowest success rate being 5.0%, it's clear that vulnerabilities are class-specific. Particularly, VGG16 exhibits the largest vulnerable regions for non-targeted attacks, averaging 41.7 pixels. However, when considering class-specific vulnerabilities, VGG16's pixel count drops to 20.6, the smallest among the three studied networks. Further analysis reveals that VGG16 has the highest diversity of target classes where a single sample can be successfully perturbed to, averaging 2.0 classes, compared to 1.7 (ResNet18) and 1.6 (NiN). This suggests their vulnerabilities are distributed across multiple target classes.

### 4.1.2    DIVERSE ADVERSARIAL EXAMPLES WITH IMAGENET

In the case of non-targeted vulnerable region discovery, we sample 500 correctly classified images. Due to the significant computational load involved in perturbing each correctly classified sample into the remaining 999 classes, we select five parent classes, each including two child classes: Bird (including snowbird, chickadee), Dog (elkhound, malamute), Plane (space shuttle, warplane), Cat (tabby, Persian), and Ship (pirate ship, schooner). Each child class contains 50 images.

**Diverse adversarial examples with non-targeted scenario**   Our algorithm is still capable of generating diverse one-pixel adversarial examples with high-resolution images, as evidenced in Table 2. Both DNNs exhibit similar success rates and pixel numbers, differing by only 1% and 0.1, respectively. Examples of vulnerable regions for high-resolution images are presented in the right three images of Fig. 4. However, it's worth noting that the average number of one-pixel perturbations accounts for only a small fraction of the total pixels in these images, making the identified vulnerable regions relatively smaller and sparser compared to those on CIFAR-10.

**Diverse adversarial examples with targeted scenario** While thousands of attacks are applied to each DNN, only four successful attacks occur for AlexNet, and fourteen are observed for ResNet50, resulting in a success rate of under 1%. Notably, all successful targeted attacks happen between pairs of child classes. These results can be

Table 2: Results on ImageNet

| Non-targeted | AlexNet | ResNet50 |
|---|---|---|
| Success rate | 11.7% | 10.7% |
| Pixel number | 20.5 | 20.6 |

attributable to several reasons: i) DNNs trained on higher-resolution ImageNet images are less vulnerable to one-pixel modifications, as evidenced in the non-targeted attack section. ii) In our experiments on CIFAR-10, we find that vulnerable regions typically only enable crossing the decision

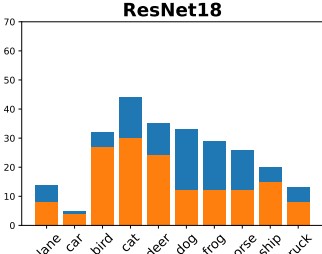 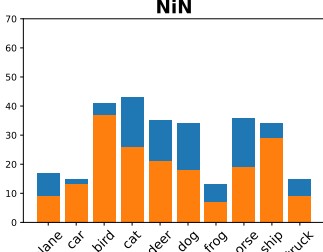 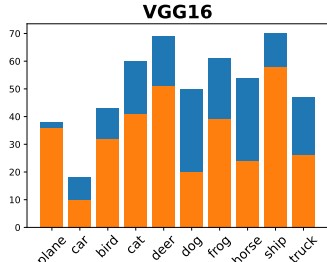

Figure 5: Number of successful attacks (vertical axis) for a specific class acting as the original class. **Orange bar:** Number of successful attacks where one-pixel perturbations are located in the image background. **Blue bar:** Number of successful attacks where one-pixel perturbations are found *only* in the foreground.

boundaries of a limited number of classes. On average, each sample can be perturbed to 1.7, 1.6, and 2.0 classes for ResNet18, NiN, and VGG16, respectively. For ImageNet models, we calculate an average of 1.04 classes for AlexNet and 1.13 classes for ResNet50 can be perturbed to, based on non-targeted attack results. Considering we select only 10 classes, there's no guarantee that the targeted vulnerable classes are among them. This reveals limitations of our algorithm when applied to high-resolution images. We plan to conduct more comprehensive studies on various types of vulnerable regions (e.g., regions with small patches in Appendix A.6) in models trained on ImageNet in future work.

## 4.2 VULNERABLE REGIONS LOCATIONS

In addition to evaluating the size of vulnerable regions, we are particularly interested in exploring whether these regions might appear in the backgrounds of images. This exploration is driven by our desire to understand the degree to

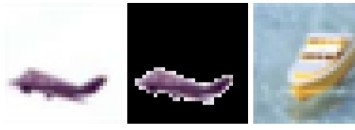 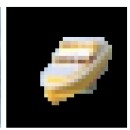

Figure 6: Examples of segmentation with Grabcut

which background elements can influence DNNs' final predictions. Experiments are conducted with CIFAR-10 for both targeted and non-targeted scenarios. Since CIFAR-10 images are low-resolution and may lack a clear demarcation between foreground and background, we utilize the Grabcut algorithm Rother et al. (2004) to segment the identified objects and background, Examples of this segmentation are provided in Fig. 6. To evaluate the influence of the location of vulnerable regions, we employ following metrics: 1) **Background Percentage:** This metric quantifies the proportion of vulnerable images where one-pixel perturbations appear in the background. 2) **BPixel Percentage:** This denotes the ratio of one-pixel perturbations that are identified in the background. 3) **Foreground/Background Effect:** This metric measures the mean confidence shift for the true label in non-targeted attacks and the mean confidence boost for a chosen false class in targeted attacks.

### 4.2.1 VULNERABLE REGIONS LOCATIONS IN NON-TARGETED SCENARIO

We re-sample 1,000 correctly classified images from the test set (100 images per class) to evaluate all three DNNs. The results, illustrating the impact of perturbation positions, are presented in Table 4.

Among the studied networks, NiN exhibits the highest dependence on vulnerable contextual information. Specifically, 67.1% of the vulnerable images for NiN are identified one-pixel perturbations in the background, which is 6.5% higher than that for ResNet18 and 1% higher than that for VGG16. Furthermore, 39.1% of all one-pixel perturbations for NiN are detected in the background, surpassing the rates for ResNet18 and VGG16 by 10.4% and 4.9%, respectively. In contrast, ResNet18 shows reduced reliance on vulnerable background features compared to the other models. However, the one-pixel perturbations in recognized objects (foreground) for ResNet18 result in more significant misclassifications: 10% more compared to NiN and 18.9% more than VGG16. This observation highlights ResNet18's increased sensitivity to changes in the foreground, suggesting the model gives higher importance to object-specific vulnerable features to achieve higher accuracy.

While one-pixel perturbations in the foreground do result in a larger drop in the confidence of the ground-truth label, it's noteworthy that perturbations in the image background still lead to a significant average reduction of 54.6% in label confidence. In fact, for over 60% of the images, we identify vulnerable regions in the background.

Table 4: Overall results on CIFAR-10 dataset. Left: Non-targeted attacks Right: Targeted attacks

| Non-Targeted | ResNet18 | NiN | VGG16 | Targeted | ResNet18 | NiN | VGG16 |
|---|---|---|---|---|---|---|---|
| Background percentage | 60.6% | 67.1% | 66.1% | Background percentage | 54.2% | 60.7% | 59.1% |
| BPixel percentage | 29.2% | 39.6% | 34.7% | BPixel percentage | 30.4% | 37.5% | 34.8% |
| Foreground effect | 74.1% | 64.1% | 55.2% | Foreground effect | 73.3% | 62.6% | 46.9% |
| Background effect | 58.3% | 58.1% | 47.3% | Background effect | 63.2% | 57.1% | 41.9% |

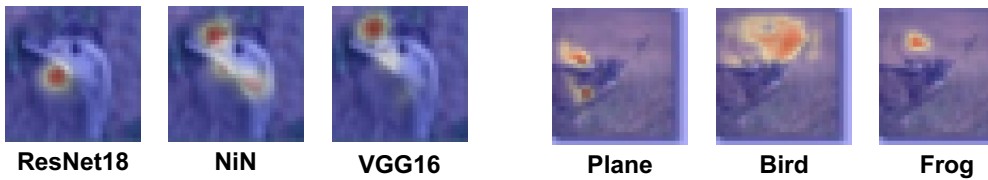

| ResNet18 | NiN | VGG16 | Plane | Bird | Frog |

Figure 7: Vulnerable regions among different DNNs(left) and targeted classes(right)

**Locations with different original classes are quite differently** The number of vulnerable images belonging to a specific class is depicted in Fig. 5. Notably, classes like planes, birds, and ships consistently exhibit a high percentage of vulnerable images with one-pixel perturbations in their backgrounds. This trend might be attributed to the simplicity of the backgrounds associated with these classes. For instance, ship images often feature the sea as a background. To explore the prevalence of certain background attributes in the training data, we randomly sample additional 1000 images (100 images per class) from the training set. We find that the attribute 'sky' appeared in 81 out of 100 sampled images belonging to the plane class. For the ship class, 62 images have the attribute of 'sky' and 90 images have the attribute of 'sea' out of the sampled 100 images. These insights imply that the vulnerability to one-pixel perturbations in classes like planes and ships might arise from the dominant presence of particular background attributes in the training data.

**Vulnerable regions are shared among different types of DNNs** We are curious to see if such vulnerable regions are shared across models given adversarial examples are usually

Table 3: Sharing Ratio Across DNN Pairs

| | Res&NiN | Res&VGG | VGG&NiN | All |
|---|---|---|---|---|
| Image ratio | 14% | 16.5% | 19.3% | 9.8% |
| Pixel ratio | 7.9% | 6.2% | 5.4% | 1.2% |

transferred across different models Inkawhich et al. (2019). The results are presented in Table 3. The 'Image Ratio' indicates the percentage of tested images that are identified as vulnerable by both DNNs listed in the first row of the table. Meanwhile, 'Pixel Ratio' signifies the average overlap of vulnerable regions (i.e., positions of one-pixel perturbations) among these shared vulnerable images. We find that only a small percentage of images are identified as vulnerable across different DNNs, with the highest overlap being 19.3% between VGG16 and NiN. The overlap in vulnerable regions within these shared vulnerable images is limited. Heatmaps for a common vulnerable image are provided in the left of Fig. 7. Additional examples can be found in Fig. 13 of Appendix A.7.

### 4.2.2 VULNERABLE REGIONS LOCATIONS IN THE TARGETED SCENARIO

In the targeted vulnerable region discovery experiment, we re-sample 500 classified images (50 images per class). Table 4 presents the ratio of one-pixel perturbations located in the image background and their corresponding effects on different DNNs. A detailed analysis of one-pixel perturbation locations across various original-target class pairs is provided in Appendix A.4.

**Some vulnerable regions are associated with multiple targeted classes** We visualize the heatmap of vulnerable regions in the right of Fig. 7, where each pixel's intensity corresponds to increase in confidence for the target classes. Additional examples are provided in Fig. 14 of Appendix A.7. With these examples, we discover that there exist common vulnerable regions across different target classes. This indicates that the samples can penetrate the decision boundaries of multiple classes along these specific dimensions. However, a detailed analysis revealed that the percentage of attacked positions, which can lead to perturbations causing more than one target classes, is only 3.0%, 1.9%, and 2.9% for ResNet18, NiN, and VGG16, respectively. While the vulnerable regions primarily concentrate on distinct areas for different target classes, there exist common vulnerable regions shared by multiple target classes.

### 4.3 Vulnerable Regions with Adversarial Training

Adversarial training is one of the most effective methods to defend against $\ell_\infty$ attacks. In this section, we evaluate the influence on CIFAR-10's vulnerable regions, focusing on two prominent adversarial training algorithms: PGD Madry et al. (2018) and TRADES Zhang et al. (2019) using ResNet18. The detailed training setup can be found in Appendix A.2. Additionally, a comprehensive analysis of the locations of one-pixel perturbations across different original-target class pairs is available in Appendix A.5.

**Adversarial training reduce the vulnerabilities along single pixels** We apply the non-targeted attack on the same sampled 1000 images for standard trained models. The results are presented in Table 5. The number of vulnerable images is significantly diminished with DNNs trained with PGD (-14.4%) and TRADES (-18.7%) algorithm. Besides the vulnerable regions getting shrunk, we can observe that sensitivity of vulnerable regions is also greatly weakened. These results provide further evidence that the $\ell_\infty$ adversarial training can smooth the loss land-scape not only within the $\ell_\infty$ ball Madry et al. (2018) but along every single dimension of images.

Table 5: Attack results with different adversarial trained models

|  | PGD | TRADES |
|---|---|---|
| Success rate | 14.0% | 9.7% |
| Pixel number | 29.0 | 17.8 |
| Background percentage | 68.9% | 65.8% |
| BPixel percentage | 49.2% | 53.6% |
| Foreground effect | 31.4% | 5.0% |
| Background effect | 20.9% | 7.6% |

**The effect of adversarial training on the location of vulnerable regions** While the success rate significantly drops, vulnerable regions are more likely to be found in the background compared to standard trained models . This suggests adversarial training may smooth loss landscape along dimensions on objects better. The impact on the output of the model gets largely reduced wherever the location of one-pixel perturbations is. Especially for the DNN

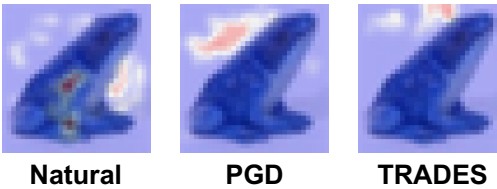

**Natural**     **PGD**     **TRADES**

Figure 8: Vulnerable regions with Standard/PGD/Trades trained models.

trained with TRADES, 69.1% and 50.7% reduction on the change of confidence of the label can be observed. An interesting phenomenon is noticed, the one-pixel perturbations located in the background have a slightly larger influence than that in the foreground for TRADES trained models. This suggests that TRADES works better on attenuating the effect of foreground one-pixel perturbations. Examples of vulnerable region heatmap for DNNs with different training algorithms are presented in Fig. 8, and more examples can be found in Fig. 15 of Appendix A.7.

### 4.4 Compare with important regions identified by the explainable approach

Explainable DNNs focus on identifying the key features that influence their predictions. In contrast, our research delves deeper, placing greater emphasis on the detection of vulnerable regions. These regions, interestingly, might not always be highlighted by standard explainability techniques. As demonstrated in Fig. 9, there's a clear difference between these vulnerable regions and the regions deemed important by the explainable DNNs. More examples of comparision can be found in Fig. 16 of Appendix A.7. It's evident that certain areas, even if overlooked as non-essential by the explainability perspective, can significantly alter the DNN's output with just minor perturbations.

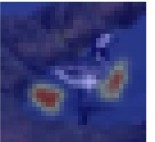 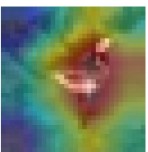

**Vulnerable**     **Important**

Figure 9: Vulnerable regions located by our algorithm and important regions with Grad-CAM.

### 5 Conclusion

In this study, we aim to reveal vulnerable regions using various adversarial examples, each with single perturbed pixels placed at different locations. Extensive experimental results demonstrates our algorithm can effectively locate large amounts of vulnerable regions, including those in the backgrounds. And valuable insights are provided with analysis of such vulnerable regions. A comprehensive study on vulnerable regions specifically for high-resolution images is expected for our future work.

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

# A APPENDIX

## A.1 RELATED WORK WITH ADVERSARIAL TRAINING:

Adversarial training, a pioneering technique introduced by Goodfellow et al. (2014), has been a cornerstone in improving the robustness of deep neural networks (DNNs) against adversarial attacks. The core concept involves training the model on adversarial examples generated by perturbing input data to maximize the loss. Madry et al. (2018) expanded on this concept by introducing the Projected Gradient Descent (PGD) adversarial training, considered one of the most effective methods for training robust models. Subsequent research has explored various dimensions of adversarial training. For instance, Xie et al. (2019) introduced feature denoising to improve model robustness. Zhang et al. (2019) proposed TRADES, a theoretical framework balancing model accuracy on clean data with robustness against adversarial examples. Furthermore, Pang et al. (2019) improved robustness by incorporating ensemble methods. In this work, we have also explored how the identified vulnerable regions change with different adversarial training algorithms.

## A.2 EXPERIMENT SETUP

**Dataset:** The CIFAR-10 dataset comprises 60,000 images, each of 32x32 dimensions, evenly distributed across 10 distinct classes with 6,000 images per class. This dataset is divided into 50,000 training images and 10,000 testing images. The ImageNet dataset, also known as ILSVRC 2012, contains high-resolution natural images spanning 1,000 classes. These images are resized to dimensions of 224×224 for DNN classification.

**Hyperparameters for diverse adversarial examples generation:** In our experimental setup, we use a Gaussian distribution represented by $\mathcal{N}(\mu = 128, \delta = 127)$ for initializing the $r, g, b$ values, whereas the coordinates $x, y$ are determined utilizing uniform distributions: $\mathcal{U}(1, 32)$ for CIFAR-10 and $\mathcal{U}(1, 224)$ for ImageNet. For the CIFAR-10 dataset, our approach uses 200 individuals, each undergoing 100 iterations using the Sharing DE procedure. Given that ImageNet images are approximately 50 times larger than those in CIFAR-10, we increase the population size to 800. The scale factor $W$ and recombination rate $R$ are predetermined at 0.5 and 1.0, respectively. Notably,

Table 6: Accumulate results of 10 runs for the one-pixel attack

| Non-targeted | ResNet18 | NiN | VGG16 | Targeted | ResNet18 | NiN | VGG16 |
|---|---|---|---|---|---|---|---|
| Success rate | 27.2% | 32.0% | 58.4% | Success rate | 5.0% | 5.7% | 12.9% |
| Pixel number | 1.3 | 1.9 | 2.7 | Pixel number | 1.2 | 1.6 | 3.2 |

a sharing radius of 4 pixels is consistently applied for both CIFAR-10 and ImageNet datasets. The parameter $\alpha$, which controls the shape of the sharing function, is set to 1.

**Training details for standard training models:** In our ImageNet experiments, we employe pretrained ResNet50 and AlexNet models directly from PyTorch. For the CIFAR-10 experiments, the models are trained using SGD with a momentum of 0.9 and a weight decay of $5 \times 10^{-4}$. We initialize the learning rate at 0.1, applying cosine annealing for its adjustment. The training process spans a total of 200 epochs.

**Training details for adversarial methods:** For both PGD and TRADES, models are trained using SGD with a momentum of 0.9 and a weight decay of $2 \times 10^{-4}$. The initial learning rate is set to 0.1 and is reduced by a factor of 10 at the 75th, 90th, and 100th epochs, respectively. A total of 120 epochs are used for training the DNN. The maximum perturbation is set at $\epsilon = 0.031$, with a step size of $\epsilon/4$ for generating adversarial examples.

## A.3 SUPERIORITY IN DISCOVERING DIVERSE ADVERSARIAL EXAMPLES

We verify the superiority of our algorithm to compare against an alternative algorithm one-pixel attack Su et al. (2019). The experiment is conducted on CIFAR-10. To generate a range of adversarial examples, we execute the one-pixel attack ten times. Results are presented in Table 6. For a fair comparison, we set the population size to 200 and limited the number of generations to 100 for this experiment.

Our algorithm identifies a comparable count of vulnerable images to the multi-run one-pixel attack Su et al. (2019), it discovers a broader variety of one-pixel perturbations with fewer computational resources. Notably, the one-pixel attack often converges to specific regions on the object across multiple runs. In contrast, our method unearths a more expansive set of vulnerable areas, highlighting its proficiency in vulnerable region discovery.

## A.4 LOCATIONS OF ONE-PIXEL PERTURBATIONS WITH ORIGINAL-TARGET PAIRS

The results in Fig. 10 illustrate the average number of one-pixel perturbations detected for each original-target class pair. Beyond findings from the previous work Su et al. (2019), we have identified additional intriguing properties. For instance, cats (class 3) are more easily perturbed to dogs and vice versa Su et al. (2019). Interestingly, the perturbations are predominantly located on the cat or dog itself, rather than the background. Additionally, we have observed that images of ships (class 8) are susceptible to being perturbed into the plane class through perturbations in the background. However, the opposite—planes being perturbed to ships—is less common. This difference may be attributed to the similar but not identical background attributes shared by these two classes. Notably, the attribute sky appears more frequently in ship images (60 out of 100), while the presence of the sea is less common in plane images (only 7 out of 100).

## A.5 THE EFFECT OF ADVERSARIAL TRAINING ON VULNERABLE PAIRS

We conduct an experiment to identify targeted vulnerable regions and observe how the average number of one-pixel perturbations in various locations changes for different original-target pairs We reuse the 500 correctly classified images for standard trained and conducted the targeted attack on the adversarial trained models. The experiment results are presented in Fig. 11. From the results, we notice a decrease in the number of diverse one-pixel perturbations for most original-target class pairs. TRADES is found to perform better in eliminating the adversarial vulnerable regions. Notably, no one-pixel perturbations are found for the original class 'plane' to other target classes. However, there are still some class pairs where the vulnerable regions increase after adversarial training. For the PGD-trained model, the class 'bird', 'cat', and 'deer' (class 2, 3, 4) are more likely to perturbed than the class 'frog' (class 6). And the class 'bird' (class 2) is more easily to perturbed than the class

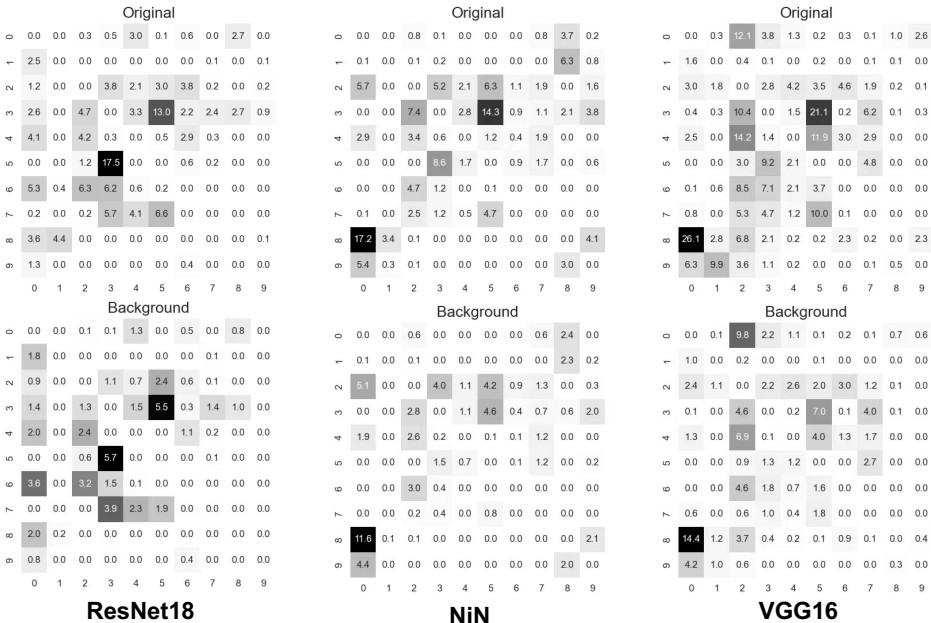

Figure 10: The average number of adversarial pixels for the original-target class pair with standard trained DNNs. Vertical and horizontal indices indicate respectively the original and target classes. Original means the average number of adversarial pixels with the whole image. Background indicates the average number of adversarial pixels on the background. The classes are identified by numbers from 0 to 9, representing the following classes respectively: plane, car, bird, cat deer, dog, frog, horse, ship, and truck.

Table 7: Results of attacks with $2 \times 2$ pixel patch with our adapted algorithm on CIFAR-10. Region size refers to the average total number of perturbed pixels covered in each successful attack.

| CIFAR-10 | Perturbation Level | Success Rate | Region Size |
|---|---|---|---|
| VGG16 (ours) | 2x2 pixel patch | 83.2% | 171.3 |
| NiN (ours) | 2x2 pixel patch | 57.4% | 122.4 |
| ResNet18 (ours) | 2x2 pixel patch | 51.2% | 105.7 |

'plane' (class 0). This suggests that while adversarial training aims to eliminate a majority of the vulnerabilities, it inadvertently introduces new vulnerabilities between other class pairs.

## A.6 ADDITIONAL EXPERIMENTS WITH DIVERSE ADVERSARIAL PATCHES:

While our proposed approach excels in identifying vulnerable regions susceptible to one-pixel perturbations, it does exhibit limitations when applied to higher-resolution images. To further assess the effectiveness of our algorithm, we conducted additional experiments by adapting it from pixel-level vulnerability assessment to patch-level vulnerability analysis. In the context of patch attacks, each candidate solution, denoted as $z_i \in \mathbb{S}$, can be modified to a tuple containing the coordinates of the top-left pixel of the patch and the RGB values of different perturbed pixels within the patch. As a result, the tuple comprises $2 + 3 \times n$ elements, where $n$ represents the number of perturbed pixels.

For our CIFAR-10 experiment, we utilized $2 \times 2$ pixel patches with 500 correctly classified images. In the case of ImageNet, where images are approximately 50 times larger than those in CIFAR-10, we employed $4 \times 4$ pixel patches with 100 correctly classified images. The results of these experiments are presented in Tables 7 and 8, where the term 'region size' refers to the average number of distinct perturbed pixels covered by diverse adversarial patches. The improved success rate and expanded region size demonstrate the adaptability of our algorithm to varying requirements. However, this modification shifts our analytical focus from individual pixels to patch-level analysis, striking a balance between achieving a higher success rate and maintaining the granularity of vulnerability assessment. We leave a comprehensive analysis for future work.

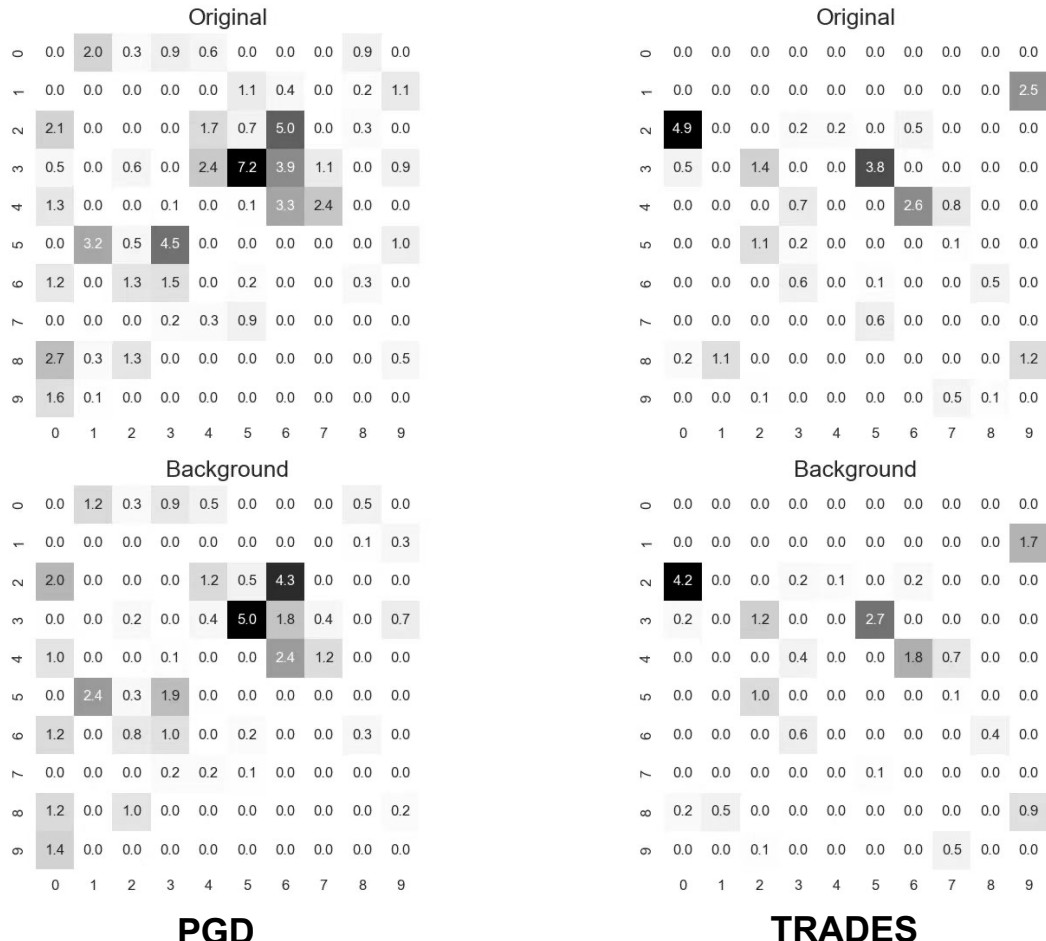

Figure 11: The average number of adversarial pixels for the original-target class pair with adversarial trained DNNs.

Table 8: Results of attacks with $4 \times 4$ patch with our adapted algorithm on ImageNet. Region size refers to the average total number of perturbed pixels covered in each successful attack.

|  | Perturbation Level | Success Rate | Region Size |
|---|---|---|---|
| AlexNet (ours) | 4x4 pixel patch | 40% | 1106.2 |
| ResNet50 (ours) | 4x4 pixel patch | 28% | 1920.5 |

## A.7 ADDITIONAL EXAMPLES OF VULNERABLE REGIONS

In this subsection, we provide extended visual illustrations to better highlight the vulnerable regions pinpointed by our algorithm. Fig. 12 depicts these areas across varying architectures for both the CIFAR10 and ImageNet datasets. Notably, these regions can identified on both recognized objects and backgrounds, independent of the image resolution. In Fig. 13 and Fig. 14, we exhibit vulnerable regions of the same images across different DNNs and different target classes, respectively,a shared vulnerability pattern. We provide more examples of vulnerable regions for adversarial trained models in Fig. 15, demonstrating that adversarial trained models are more likely to have vulnerable regions in the background. Finally, we present more examples to compare the vulnerable regions discovered by our algorithm with important regions identified by two explainable DNN methods by Grad-CAM Selvaraju et al. (2017) and Full-Grad methods Srinivas & Fleuret (2019) in Fig. 16. This comparison highlights that certain regions, often missed by explainability tools, can significantly influence the DNN's predictions with slight changes.

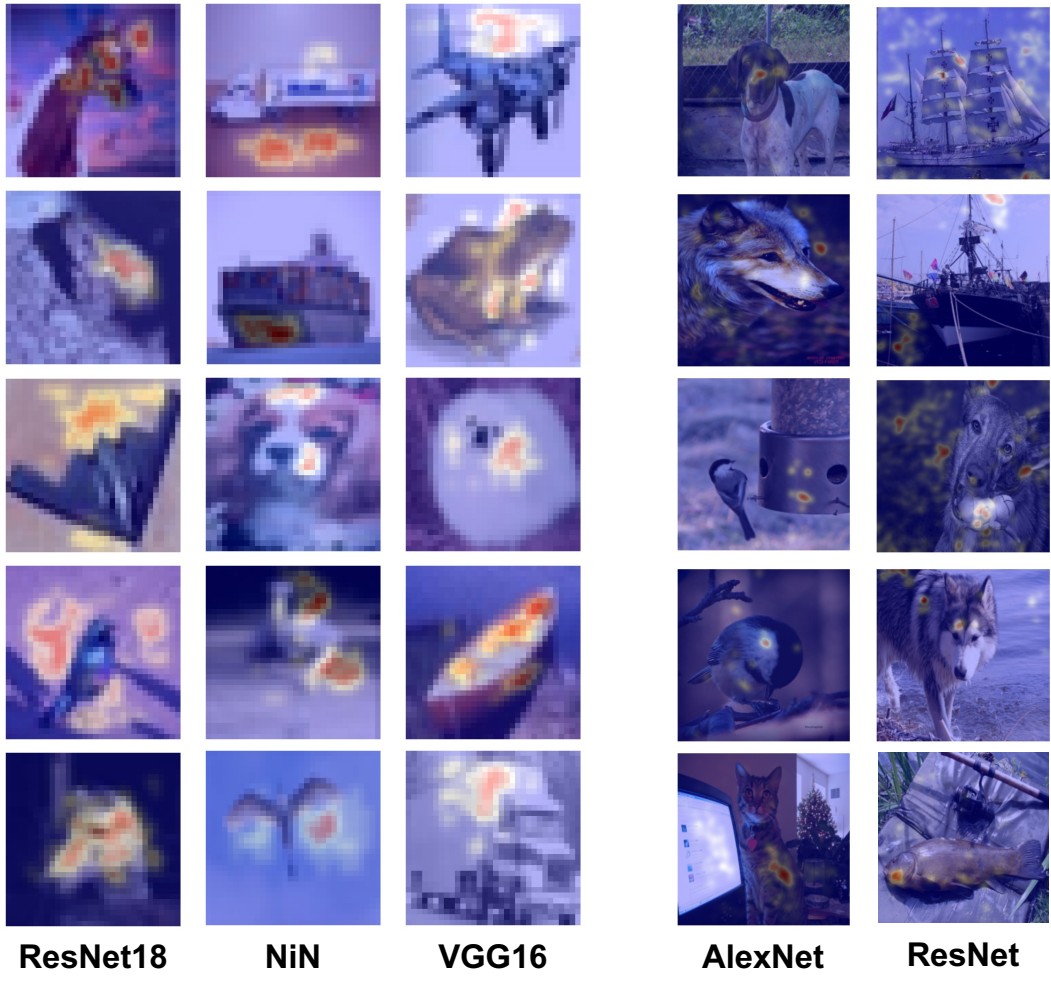

**ResNet18**     **NiN**     **VGG16**     **AlexNet**     **ResNet**

Figure 12: Visualization of vulnerable regions with CIFAR10(left) and ImageNet (right).

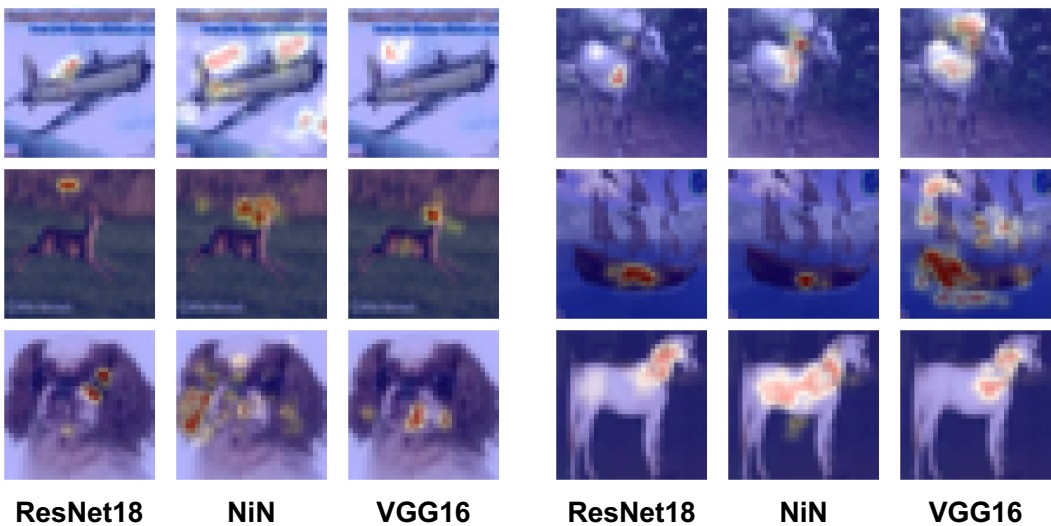

**ResNet18**     **NiN**     **VGG16**     **ResNet18**     **NiN**     **VGG16**

Figure 13: Visualization of vulnerable regions for shared vulnerable images by 3 different DNNs.

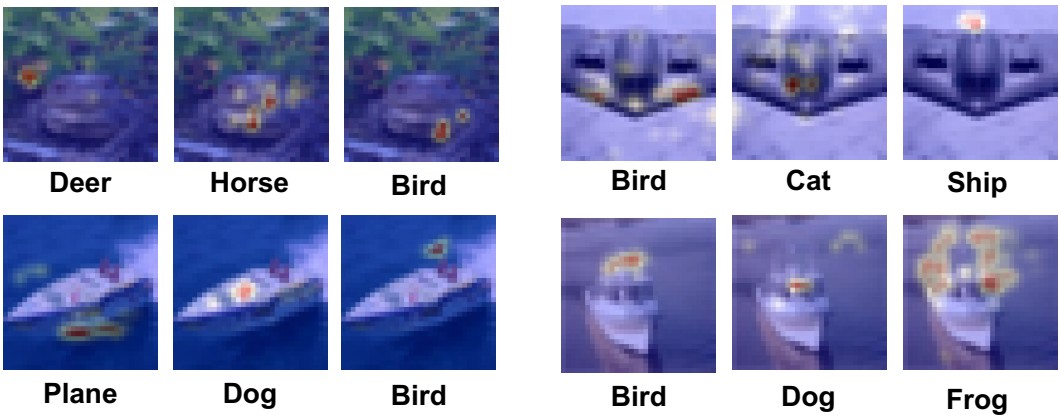

Figure 14: Visualization of vulnerable regions to different false classes.

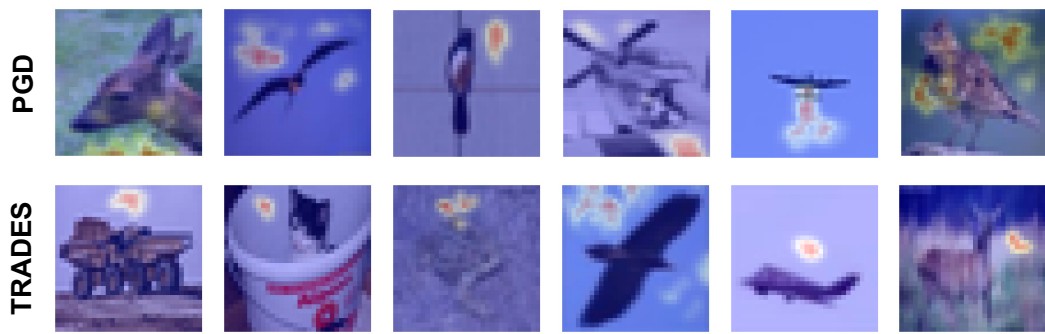

Figure 15: Vulnerable regions of adversarial trained ResNet18 Models.

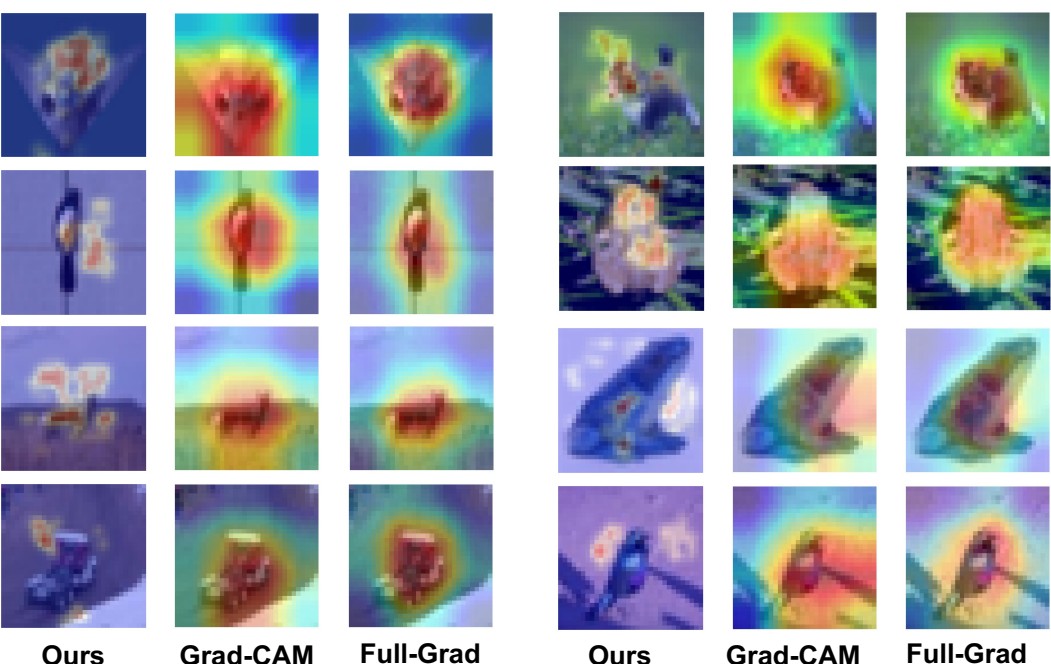

Figure 16: Vulnerable regions discovered by our algorithm and Important regions discovered by Grad-CAM Selvaraju et al. (2017) and Full-Grad methods Srinivas & Fleuret (2019).

