# OpenReview forum: "Vulnerable Region Discovery through Diverse Adversarial Examples"
_ICLR.cc/2024/Conference — Submitted to ICLR 2024_

### Official Review · Reviewer_DTs6 · 2023-10-28

**Soundness:** 4 excellent
**Presentation:** 3 good
**Contribution:** 3 good
**Rating:** 6
**Confidence:** 4

**Summary:**

This paper utilize the Sharing Differential Evolution algorithm to come up with multiple signle-pixel adversarial attacks at diverse locations. The attack region reveals the vulnerable region in the input image.

**Strengths:**

1. The idea of utlizing single-pixel adversarial attack to highlight vulnerable region is interesting
2. The paper is overall well written and easy to follow
3. The paper make interesting observation on how adversarial training affect vulnerable region, and how teh region changes with different source/target classes

**Weaknesses:**

My major concern of the paper is the lack of technical contribution. As an attack, the proposed method is not effectively leading to high success rate, and is not well bounded (e.g. constraining the maximum number of pixels to be perturbed etc.) While as a visualization/explaination method, it is not well motivated/explained how the identification of vulnerable region can help better understand or improve the model. This hinders the significance of the paper.

**Questions:**

See weakness

---

> ### Author Response · Authors · 2023-11-19
> **Response to Reviewer DTs6 (part 1)**
>
> We greatly appreciate your insightful and constructive feedback. We carefully address your concerns in the following discussion.
> ___
>
> >**Question:** My major concern of the paper is the lack of technical contribution. As an attack, the proposed method is not effectively leading to high success rate, and is not well bounded (e.g. constraining the maximum number of pixels to be perturbed etc.) While as a visualization/explaination method, it is not well motivated/explained how the identification of vulnerable region can help better understand or improve the model. This hinders the significance of the paper.
>
> **Answer:** We genuinely appreciate the reviewer’s insightful question. We give the following explanation for addressing your concerns.
>
> **Clarify why our algorithm is bounded with one pixel:**
>
> Our study employs diverse one-pixel perturbations, a deliberate choice aimed at meticulously uncovering vulnerable regions in deep neural networks. It seeks to enhance the comprehensibility of these networks by demonstrating how single-pixel changes can expose significant vulnerabilities. Our algorithm is specifically designed for detailed, pixel-level analysis, highlighting the impact of even the smallest modifications on deep models. This constraint to one-pixel perturbations is **not just a limitation but a strategic method to reveal the nuanced sensitivity and potential weaknesses** within these complex systems.

---

> ### Author Response · Authors · 2023-11-19
> **Response to Reviewer DTs6 (part 2)**
>
> **Clarify effectiveness as an attack algorithm:**
>
> 1. Our algorithm, designed as an attack method, proves effective in this one-pixel attack context. We have benchmarked our approach against the existing one-pixel attack [1] on the CIFAR-10 dataset with below Table 3, as detailed in Appendix A.2. In terms of success rates, our algorithm performs competitively with the conventional one-pixel attack. However, a key distinction is that our method identifies a greater number of effective one-pixel perturbations across diverse locations, demonstrating its superior capability in revealing the vulnerabilities of DNNs.   The enhanced ability to uncover vulnerabilities is particularly valuable in safety-critical scenarios, such as autonomous driving and image-based medical diagnosis.
>
> 2. The success rate can be further improved by allowing more perturbed pixels. This adaptation necessitates the use of spatial constraints to accurately represent vulnerabilities in specific areas, such as using small image patches. For patch attacks, we redefine each candidate solution as a tuple. This tuple includes the coordinates of the patch's top-left pixel and the RGB values for the perturbed pixels within the patch, resulting in a tuple of $2 +3 \times n$ elements, where $n$ is the number of perturbed pixels. We have added this experiment (in Appendix A.6): $2\times 2$ pixel patch attacks on the CIFAR-10 dataset and $4\times 4$ pixel patch attacks on the ImageNet dataset. In the CIFAR-10 experiments, we randomly selected 500 correctly classified images, each being 32x32 pixels. In the ImageNet experiments, we used 100 randomly chosen, correctly classified images, each resized to 224x224 pixels. We have already included this experiment in Apendix A.6 of the revised manuscirpt.
>
>     The outcomes of these experiments are detailed in Tables 1 and 2.  In these tables, a cost of 1x corresponds to 20,000 queries to DNNs. In the CIFAR-10 experiments, we observed a significant improvement in both the success rate and the size of the discovered vulnerable regions with this adjustment. A similar trend was noted in the ImageNet experiments. For example, using AlexNet, our analysis identified 95.2 distinct patches, covering 1106.2 pixels in total. Notably, $1106.2\neq95.2×16$, which suggests overlaps among the patches. In contrast, with ResNet50, which had a lower success rate, we found 174.8 different patches covering 1920.5 pixels. These findings highlight the effectiveness and adaptability of our approach.  We leave a detailed analysis of such patch level vulnerability analysis as our future work.
>
> **Tabel1.** Results of 10 runs for the one-pixel attack, a single run of our algorithm with one-pixel perturbation and a single run of our algorithm with 2x2 pixel patch on CIFAR-10. Region size refers to the average total number of perturbed pixels covered in each successful attack.
> |            | Perturbation Level | Success Rate | Region Size | Cost |
> |------------|--------------------|--------------|-------------|------|
> | VGG16 [1]  | one pixel          | 58.4%        | 2.7         | 10x  |
> | NiN [1]    | one pixel          | 32.0%        | 1.9         | 10x  |
> | ResNet18 [1]| one pixel         | 27.2%        | 1.3         | 10x  |
> |------------|--------------------|--------------|-------------|------|
> | VGG16 (ours)| one pixel         | 60.2%        | 41.7        | 1x   |
> | NiN (ours)  | one pixel         | 31.6%        | 40.8        | 1x   |
> | ResNet18 (ours)| one pixel      | 28.4%        | 31.3        | 1x   |
> |------------|--------------------|--------------|-------------|------|
> | VGG16 (ours)| 2x2 pixel patch   | 83.2%        | 171.3       | 1x   |
> | NiN (ours) | 2x2 pixel patch    | 57.4%        | 122.4       | 1x   |
> | ResNet18 (ours)| 2x2 pixel patch| 51.2%        | 105.7       | 1x   |
>
> **Tabel2.** Comparison of vulnerable region discovery using one pixel and 4x4 pixel adversarial patches on ImageNet with our algorithm. Region size refers to the average total number of perturbed pixels covered in each successful attack.
> |                   | Perturbation Level | Success Rate | Region Size |
> |-------------------|--------------------|--------------|-------------|
> | AlexNet (ours)    | one pixel          | 11.7%        | 20.5        |
> | ResNet50 (ours)   | one pixel          | 10.7%        | 20.6        |
> | AlexNet (ours)    | 4x4 pixel patch    | 40%          | 1106.2      |
> | ResNet50 (ours)   | 4x4 pixel patch    | 28%          | 1920.5      |
>
> [1] Jiawei Su, Danilo Vasconcellos Vargas, and Kouichi Sakurai. One pixel attack for fooling deep
> neural networks. IEEE Transactions on Evolutionary Computation, 23(5):828–841, 2019. doi:
> 10.1109/TEVC.2019.2890858.

---

> ### Author Response · Authors · 2023-11-19
> **Response to Reviewer DTs6 (part 3)**
>
> **Clarify the motivation of our algorithm:**
>
> **The motivation behind our work is to advance the field of explainable deep neural networks (DNNs) from another perspective - vulnerability by adopting a novel approach.** Traditional methods typically focus on identifying the most relevant parts of images that contribute to a model's final predictions. In contrast, our algorithm aims to pinpoint regions that are particularly sensitive to subtle perturbations. Recognizing these vulnerable regions is crucial for understanding the reliability of the features acquired by the model,  and for guiding subsequent improvements.
>
> **Clarifty how the proposed algorithm can help better understand the model:**
>
> 1. By understanding the specific conditions under which a DNN fails, we can better comprehend its limitations. For instance, our algorithm found that vulnerable regions are more likely to appear in the background of the classes 'plane' and 'ship' (as discussed in **Section 4.2.1**).  This finding emphasizes the significant role of background information in the model's decision-making process. Such an observation can be unexpected and, more importantly, poses a particular risk for these two classes. In this case, the emphasis on background regions highlights the need for further model refinement or additional data preprocessing to mitigate the risk associated with such vulnerabilities).
>
> 2. In addition, our algorithm has uncovered intriguing insights into the nature of vulnerable regions. Notably, we observed that vulnerable regions can exhibit overlap across different neural networks (as discussed in **Section 4.2.1**) and can also be shared among different target classes (as discussed in **Section 4.2.2**) for the same neural networks. These findings carry important implications for understanding the vulnerabilities of DNNs. The former one suggests the potential for transferability of one-pixel perturbed adversarial examples. The latter finding demonstrates that certain image regions are particularly susceptible to perturbations, leading to varying predictions by DNNs, which underscores their significance. These findings have implications for cross-model adversarial defense and deeper investigations into the functioning of neural networks, ultimately advancing our knowledge of their capabilities and vulnerabilities.
>
> **Clarify the potential application of our algorithm to enhance robustness:**
>
> In this work, our proposed approach focused on identifying such vulnerable regions. Based on knowledge of vulnerable regions, we give the potential strategies for further robustness enhancement:
>
> - Adversarial training: We can strengthen models by employing adversarial examples that incorporate perturbations strategically crafted based on identified vulnerable regions, thereby boosting the effectiveness of adversarial training.
> - Ensemble Methods: We can enhance the robustness of deep ensembles by merging various models or defense techniques which are designed to address vulnerabilities in diverse vulnerable regions.
> - Input Preprocessing: We can apply preprocessing techniques to alter data before it enters the model, reducing the impact of potential perturbations in these high-risk, vulnerable regions.
>
> While adversarial training, ensemble methods, and input preprocessing are well-recognized for enhancing deep model robustness, integrating insights into vulnerable regions enables more **precise and targeted** defenses.

---

> > ### Comment · Reviewer_DTs6 · 2023-11-21
> > **Thank you for your response. Some clarification needed**
> >
> > I would like to thank the author for the detailed response. Here are some additional clarifications needed to better understand the merit of this paper.
> >
> > 1. Regarding the effectiveness you discussed in Table 1 of your reply, it is unclear to me how the region size is related to the attack success rate. Is it every pixel in the region can lead to such a success rate? Or is it all pixels combined in the region leads to a success attack? This is related to my comment on "not well bounded" as the proposed method does not effectively control the final region size of the attack.
> > 2. As the author has provided some evidence that the identified region is vulnerable, I don't think much evidence is available that other regions of the image is less vulnerable or not vulnerable. Some easy way to verify this would be comparing the adversarial attack/random noise performance of only modifying the vulnerable region vs. only modifying other regions.
> > 3. For the "potential application" proposed by the author, it is unclear why having the knowledge of a potential vulnerable region will help on these applications, as all these applications seems to be able to done easily with the whole image. Some emperical results are welcomed to show how the region identified by the proposed method helps in these cases.

---

> ### Author Response · Authors · 2023-11-22
> **Thank you for your response. More clarification (part 1)**
>
> Thank you very much for your insightful and constructive feedback. To address your concerns, we offer the following explanations:
> ___
> >**Question1:** Regarding the effectiveness you discussed in Table 1 of your reply, it is unclear to me how the region size is related to the attack success rate. Is it every pixel in the region can lead to such a success rate? Or is it all pixels combined in the region leads to a success attack? This is related to my comment on "not well bounded" as the proposed method does not effectively control the final region size of the attack.
>
> **Answer:**
>
> **Clarify`Is it every pixel in the region can lead to such a success rate? Or is it all pixels combined in the region leads to a success attack?':**
>
> It is right that 'Every pixel in the region can lead to such a success attack' in the perturbation level of **one-pixel**. In our additional experiments involving **patch-level** attacks, each small patch of a predefined size within the region can lead to a successful attack. For instance, in our additional experiments on CIFAR-10 presented in Table 1, each 2x2 patch composed of 4 perturbed pixels within the identified vulnerable regions leads to a successful attack.
>
>  Regarding success rates, our algorithm achieves performance comparable to existing one-pixel attack methods [1], but with the added capability of generating a much wider diversity of one-pixel perturbations. These two metrics—success rate and diversity of perturbations—position our approach as a valuable method for explaining and enhancing the interpretability of deep models, particularly from the perspective of their vulnerabilities.
>
> **Clarify `This is related to my comment on "not well bounded" as the proposed method does not effectively control the final region size of the attack.'**
>
> In response to your observation regarding the "not well bounded" aspect of our approach, it's important to clarify that the size of the region is, in fact, bounded by the population size, a key hyperparameter in our evolutionary algorithm.
>
> [1] Jiawei Su, Danilo Vasconcellos Vargas, and Kouichi Sakurai. One pixel attack for fooling deep
> neural networks. IEEE Transactions on Evolutionary Computation, 23(5):828–841, 2019. doi:
> 10.1109/TEVC.2019.2890858.
>
> >**Question2:**    As the author has provided some evidence that the identified region is vulnerable, I don't think much evidence is available that other regions of the image is less vulnerable or not vulnerable. Some easy way to verify this would be comparing the adversarial attack/random noise performance of only modifying the vulnerable region vs. only modifying other regions.
>
> **Answer:**
>
>  While it is true that our focus has been on demonstrating the vulnerability of the identified region, we believe there is substantial implicit evidence supporting the relative robustness of other image regions. **Throughout the iterative process of our evolutionary algorithm, attacks were not limited to the identified vulnerable regions; they also encompassed other pixels across the image.** However, perturbations applied to regions outside the identified vulnerable areas consistently failed to deceive DNNs. These instances were systematically excluded from our final results.
>
> This process inherently indicates that pixels or small patches outside the vulnerable regions are less susceptible to causing prediction changes when perturbed. The nature of the evolutionary algorithm ensures a comprehensive exploration of the image space, including both the final identified vulnerable areas and other regions. The fact that only specific regions repeatedly emerged as vulnerable, while others did not, implicitly supports the notion of their relative robustness. The current findings, derived from our evolutionary algorithm's exploration, already suggest a clear disparity in vulnerability across different image regions.

---

> ### Author Response · Authors · 2023-11-22
> **Thank you for your response. More clarification (part 2)**
>
> >**Question3:** For the "potential application" proposed by the author, it is unclear why having the knowledge of a potential vulnerable region will help on these applications, as all these applications seems to be able to done easily with the whole image. Some emperical results are welcomed to show how the region identified by the proposed method helps in these cases.
>
> **Answer:** Thank you for your observation regarding the potential applications mentioned in our work. Our primary contribution lies in identifying vulnerable regions within deep neural networks and providing detailed analysis and insights based on these identified areas. While the utilization of these vulnerable regions to improve the robustness of deep models is indeed relevant, it falls outside the immediate scope of our current study. Therefore, our discussion primarily addresses the concern that 'all these applications could potentially be executed using the entire image.' We leave the algorithm that can further used to enhance robustness as our future work:
>
> - Adversarial Training: The primary concept of adversarial training is to include adversarial examples in the training process to strengthen the robustness of deep models ([1, 2]). Conventionally, these adversarial examples, bound by the $\ell_\infty$ norm, are applied across the entire image.  However, as detailed in `Section 4.3' of our study, this approach does not fully eliminate vulnerable regions in models trained via this method. To address this, adversarial training that takes into account the knowledge of vulnerable regions could be a valuable complement to current techniques. This method would involve fine-tuning adversarially trained models by selectively applying larger perturbations to pixels within these identified vulnerable regions, thereby augmenting robustness. Applying larger perturbations uniformly across the entire image is not feasible, as it would render the image unrecognizable.
>
>
> - Ensemble Methods: By identifying a variety of vulnerable regions, we have the opportunity to customize different models within the ensemble to target specific vulnerable pixel. With each model in the ensemble concentrating on defending against distinct vulnerable pixels, the ensemble as a whole benefits from the combined strengths of its members. This synergy within the ensemble presents an opportunity to develop a more robust and comprehensive defense strategy overall.
>
> - Input Preprocessing: Input preprocessing ([3, 4]), such as Gaussian smoothing, JPEG compression, and quantization, have been recognized as important techniques to defend against adversarial attacks. However, applying these preprocessing methods uniformly across the entire image can often lead to the loss of critical details in recognized features. This is where the knowledge of specific vulnerable regions becomes particularly valuable. 2. By pinpointing these vulnerable regions, we can apply preprocessing techniques selectively, thereby preserving the integrity and details of the non-vulnerable parts of the image. This targeted approach aims to not only maintain the overall image quality but also enhance the efficiency of the defense mechanism.
>
> In summary, our core contribution lies in identifying vulnerable regions within deep neural networks and offering detailed analysis and insights based on these regions. Additionally, we have discussed the potential advantages of incorporating knowledge about these vulnerable regions into targeted defense strategies.
>
> [1] Aleksander Madry, Aleksandar Makelov, Ludwig Schmidt, Dimitris Tsipras, and Adrian Vladu. To-
> wards deep learning models resistant to adversarial attacks. In International Conference on Learn- ing
> Representations, 2018
>
> [2] Hongyang Zhang, Yaodong Yu, Jiantao Jiao, Eric Xing, Laurent El Ghaoui, and Michael Jordan.
> Theoretically principled trade-off between robustness and accuracy. In International Conference on
> Machine Learning, pp. 7472–7482. PMLR, 2019.
>
> [3] Naseer M, Khan S, Porikli F. Local gradients smoothing: Defense against localized adversarial attacks. In IEEE Winter Conference on Applications of Computer Vision. IEEE, pp. 1300-1307 2019
>
> [4] C. Guo, M. Rana, M. Cisse, and L. van der Maaten. Countering adversarial images using input transformations. In International Conference on Learning Representations, 2017

---

> > ### Comment · Reviewer_DTs6 · 2023-11-22
> >
> > I would like to thank the author for the extensive responses. Now I have a better understanding of the contribution of this submission, and I'm convinced of the merit of this work. I'm increasing my rating to weak accept.

---

> > > ### Author Response · Authors · 2023-11-22
> > >
> > > Thank you for your positive feedback and constructive insights regarding our work.  We are grateful for your thoughtful review and the positive impact it has had on refining and improving our study. Your thoughtful review has been instrumental in refining and enhancing our study, and we are truly grateful for the significant impact it has made.

---

### Official Review · Reviewer_rfSA · 2023-10-30

**Soundness:** 4 excellent
**Presentation:** 3 good
**Contribution:** 4 excellent
**Rating:** 8
**Confidence:** 5

**Summary:**

The paper introduces a novel approach to uncover vulnerable regions using
one-pixel perturbations located at various positions. Extensive experiments
involving various network architectures and adversarial training models have
been conducted, demonstrating that the proposed algorithm can indeed discover
diverse adversarial perturbations. Additionally, a large number of well-designed
experiments are conducted to study the properties of discovered regions, which
provide some interesting insights to deep models.

**Strengths:**

**1.** It is quite an interesting work. This paper provides a new perspective on
understanding the weaknesses of DNNs with diversely located perturbations. Specifically, this work focuses on pinpointing vulnerable regions to
single-pixel perturbations, distinguishing itself from interpretable methods
emphasizing significant areas influencing the network’s final output. This
approach facilitates the identification of specific vulnerable image areas
deserving more attention.

**2.** The paper conducts comprehensive experiments involving a range of network architectures and adversarial training techniques, clearly showcasing
how this method aids in assessing model vulnerability and contributes to
improving model interpretability. Experiment results demonstrate that
the proposed approach can effectively generate diverse adversarial perturbations to form vulnerable regions.

**3.** Beyond vulnerability detection, the paper also conducts a series of well-designed experiments to study the properties of the discovered regions,
e.g., how adversarial training influences the position of such vulnerable
regions. Overall, this approach leads to valuable insights into the behavior
of deep models, contributing to a deeper understanding of DNNs.

**Weaknesses:**

**1.** The proposed approach exhibits limitations when applied to high-resolution
images. Even though the author increased the population size to 800 for
high-resolution images, this still only represents approximately 1.5% of the
total images. However, it requires up to 80,000 queries to deep models.

**2.**  It might be better if the author could briefly introduce adversarial training
in related works section.

**3.** While the authors provide details on acquiring adversarial-trained models,
information on obtaining standard-trained models is lacking.

**Questions:**

This work provides a novel way to understand the weaknesses of DNNs, and
includes some well-designed, intriguing experiments. I have the following questions:

**1.** The experimental results reveal limitations when applied to high-resolution
images. On average, only around 20 diverse adversarial examples are found
for what the author refers to as vulnerable images. This number is relatively small. Is it possible to further adapt the method to high-resolution
images?

**2.** While understanding vulnerabilities is crucial, the paper may benefit from
discussing potential real-world scenarios where such vulnerabilities can be
exploited and their implications

---

> ### Author Response · Authors · 2023-11-19
> **Response to Reviewer rfSA (part 1)**
>
> We greatly appreciate your insightful and constructive feedback. We address your concerns in the following discussion.
> ___
>
> >**Question1: The proposed approach exhibits limitations when applied to high-resolution images. Even though the author increased the population size to 800 for high-resolution images, this still only represents approximately 1.5\% of the total images. However, it requires up to 80,000 queries to deep models.**
>
> **Answer:** We genuinely appreciate the reviewer's insightful question regarding the limitations of dealing with high-resolution images.
> The current implementation indeed only samples a small fraction of the total image population, even with an increased sample size. The necessity of up to 80,000 queries to deep models for these images is a significant demand on computational resources. However, the allowance of more localized perturbed pixels can be utilized (e.g., small patches) to increase identified vulnerable region size , thereby enhancing  the effective of our algorithm for higher resolution images. Although, such action may sacrifice the granularity of vulnerability analysis.  There is an inherent trade-off between detailed analysis of vulnerability and the the rate of successful attacks.  We recognize this as an area for improvement and leave it as our future work.
>
> ___
>
> >**Question2: It might be better if the author could briefly introduce adversarial training in related works section.**
>
> **Answer:** We are grateful for you suggestion,  We have included related works on adversarial training in **Appendix A.1**.
>
> Adversarial training, a pioneering technique introduced by Goodfellow et al. (2014) [1], has been a cornerstone in improving the robustness of deep neural networks (DNNs) against adversarial attacks. The core concept involves training the model on adversarial examples generated by perturbing input data to maximize the loss. Madry et al. (2018) [2] expanded on this concept by introducing the Projected Gradient Descent (PGD) adversarial training, considered one of the most effective methods for training robust models. Subsequent research has explored various dimensions of adversarial training. For instance, Xie et al. (2019) [3] introduced feature denoising to improve model robustness. Meanwhile, Zhang et al. (2019) [4] proposed TRADES, a theoretical framework balancing model accuracy on clean data with robustness against adversarial examples. Furthermore, Pang et al. (2019) [5] improved robustness by incorporating ensemble methods. In this work, we also explored how the identified vulnerable regions change with different adversarial training algorithms.
>
> [1]Ian J. Goodfellow, Jonathon Shlens, and Christian Szegedy. Explaining and Harnessing Adversarial
> Examples. arXiv e-prints, art. arXiv:1412.6572, December 2014.
>
> [2] Aleksander Madry, Aleksandar Makelov, Ludwig Schmidt, Dimitris Tsipras, and Adrian Vladu. To-
> wards deep learning models resistant to adversarial attacks. In International Conference on Learn-
> ing Representations, 2018.
>
> [3] Cihang Xie, Yuxin Wu, Laurens van der Maaten, Alan L Yuille, and Kaiming He. Feature denoising
> for improving adversarial robustness. In Proceedings of the IEEE/CVF conference on computer
> vision and pattern recognition, pp. 501–509, 2019
>
> [4] Hongyang Zhang, Yaodong Yu, Jiantao Jiao, Eric Xing, Laurent El Ghaoui, and Michael Jordan.
> Theoretically principled trade-off between robustness and accuracy. In International Conference
> on Machine Learning, pp. 7472–7482. PMLR, 2019.
>
> [5] Tianyu Pang, Kun Xu, Chao Du, Ning Chen, and Jun Zhu. Improving adversarial robustness via
> promoting ensemble diversity. In International Conference on Machine Learning, pp. 4970–4979.

---

> ### Author Response · Authors · 2023-11-19
> **Response to Reviewer rfSA (part 2)**
>
> >**Question3: While the authors provide details on acquiring adversarial-trained models, information on obtaining standard-trained models is lacking.**
>
> **Answer:** Thank you for your suggestion. We have added information with standard trained models in **Appendix A.2**.
>
> In our ImageNet experiments, we employed pretrained ResNet50 and AlexNet models directly from PyTorch. For the CIFAR-10 experiments, the models were trained using SGD with a momentum of 0.9 and a weight decay of $5\times 10^{-4}$
>  . We initialized the learning rate at 0.1, applying cosine annealing for its adjustment. The training process spanned a total of 200 epochs.
> ___
> >**Question4: The experimental results reveal limitations when applied to high-resolution images. On average, only around 20 diverse adversarial examples are found for what the author refers to as vulnerable images. This number is relatively small. Is it possible to further adapt the method to high-resolution images?**
>
> **Answer:** We genuinely appreciate the reviewer's insightful question and offer the following explanations on adapting our algorithm to higher-resolution images with minor adjustments.
>
> **From the pixel-level attack to the patch-level attack:** Our algorithm is adaptable to higher-resolution images by adjusting the number of pixels subject to perturbation. In these scenarios, it's crucial to apply spatial constraints that accurately reflect vulnerabilities in specific areas, like small patches within an image. In the context of patch attacks, the only modification is to the representations of candidate solutions. Each candidate solution can be modified to a tuple containing the coordinates of the top-left pixel of the patch and the RGB values of different perturbed pixels within the patch. As a result, the tuple comprises $2+3\times n$ elements, where $n$ represents the number of perturbed pixels.
>
> We have tested this approach using the ImageNet dataset with 4x4 pixel patches , selecting 100 correctly classified images resized to 224x224 pixels. Results are presented in Tabel 1. For AlexNet, our analysis identified 95.2 distinct patches, collectively covering 1106.2 pixels. It is important to note that the size of the region does not equate to the product of the number of diverse patches and the size of each patch (e.g., $1106.2 \neq 95.2 \times 16$), which indicates overlapping among the patches. In the case of ResNet50, which exhibits a lower success rate compared to AlexNet, we found 174.8 different patches that cover a total of 1920.5 pixels.   These experiment results demonstrate that our algorithm can be effectively adapted for high-resolution images with minimal adjustments. We leave a detailed analysis for high resolution image for our future work.
>
> **Tabel 1.** Comparison of vulnerable region discovery using one pixel and 4x4 pixel adversarial patches on ImageNet with our algorithm. Region size refers to the average total number of perturbed pixels covered in each successful attack.
> |                    |Perturbation Level | Success Rate | Region Size |
> |-------------------|--------------------|--------------|-------------|
> | AlexNet (ours)    | one pixel          | 11.7%        | 20.5        |
> | ResNet50 (ours)   | one pixel          | 10.7%        | 20.6        |
> | AlexNet (ours)    | 4x4 pixel patch    | 40%          | 1106.2      |
> | ResNet50 (ours)   | 4x4 pixel patch    | 28%          | 1920.5      |

---

> ### Author Response · Authors · 2023-11-19
> **Response to Reviewer rfSA (part 3)**
>
> >**Question5: While understanding vulnerabilities is crucial, the paper may benefit from discussing potential real-world scenarios where such vulnerabilities can be exploited and their implications.**
>
> **Answer:** We genuinely appreciate the reviewer's insightful question regarding the potential applications and their implications of our work. We give the following disscussion about the potential application of our proposed approach:
>
> - **Enhance the Understanding of DNN's Vulnerability:** By examining the regions of an image that, when altered, result in misclassification, researchers can infer the features a DNN relies on for decision-making and, crucially, assess the reliability of these features. Additionally, gaining insight into scenarios where a DNN fails allows for a deeper understanding of its limitations.
>
> - **Enhance the Robustness of Deep Models:** Vulnerable regions are specific areas in the input space sensitive to minor disturbances that can drastically alter deep neural network (DNN) predictions. Acknowledging these regions is vital for fortifying the model's defense against adversarial attacks and mitigating unpredictable behaviors, including:
>     - Adversarial Training: We can strengthen models by using adversarial examples carefully crafted for these identified sensitive areas
>       to enhance the efficacy of adversarial training.
>     - Ensemble Methods: We can combine multiple models or defenses to create an ensemble that collectively addresses vulnerabilities
>        in different regions for enhancing robustness of deep ensembles.
>     - Input Preprocessing: We can implement preprocessing techniques to modify images before they enter the model It can diminish the
>       effects of potential perturbations in these high-risk vulnerable regions.
>
> These approaches of detecting and correcting model vulnerabilities can be utilized to enable DNNs to preserve their performance stability when faced with adversarial attacks.

---

> > ### Comment · Reviewer_rfSA · 2023-11-22
> > **Thanks for the rebuttal.**
> >
> > Thank you very much for your comprehensive response, which has successfully addressed my concerns.
> >
> > The utilization of diverse adversarial examples to enhance the interpretability of deep models in your work is commendable, especially considering the current trend of focusing primarily on improving the robustness of deep models through explainable methods. The well-crafted experiments and their insightful findings have revealed several fascinating aspects of vulnerable regions, potentially contributing significantly to the research on adversarial vulnerabilities in deep neural networks.
> >
> > After carefully reading the response and other reviewers’ comments, I tend to maintain my original score (8: good paper).

---

> > > ### Author Response · Authors · 2023-11-22
> > >
> > > Thank you for your encouraging remarks and recognition of our work. We sincerely appreciate your affirmative feedback and the acknowledgment of our research efforts. Your constructive insights have been invaluable in guiding the refinement and progression of our study.

---

### Official Review · Reviewer_4smo · 2023-10-30

**Soundness:** 3 good
**Presentation:** 2 fair
**Contribution:** 2 fair
**Rating:** 5
**Confidence:** 3

**Summary:**

A one-pixel adversarial attack is an alteration to a single pixel that changes the network's prediction on an image.  If you take an image and repeatedly existing run one-pixel attacks, most of these runs will change the same pixel or small set of pixels.  In this paper, the authors are interested in identifying a large set of pixels that _each_ can be modified in such a way that fully changes the network's prediction.  To that end, the paper proposes an evolutionary algorithm that takes as input an image and a network, and returns a _set of distinct_ one pixel attacks that all succeed.  Often, these pixels are spatially clustered together, so the authors refer to this set of pixels as a "vulnerable region."  In section 4.4 authors argue that finding 'vulnerable regions' of an image can be one way to interpret a neural network.  (It seems to sometimes highlight very different regions than GradCAM.) The authors apply their method for finding vulnerable regions on both CIFAR-10 and ImageNet for both untargeted and targeted perturbations.

**Strengths:**

Originality: the paper is original, as I'm not aware of other methods for simultaneously finding distinct sets of one-pixel adversarial attacks

Quality: the paper is of reasonable quality

Clarity: most of the paper was clear, but I thought the abstract was confusing: "Traditional norm-based adversarial example generation algorithms, due to their lack of spatial constraints, often distribute adversarial perturbations throughout images, making it hard to identify these specific vulnerable regions." If I understand correctly, the proposed method does not have spatial constraints that encourage the found pixels to be close together.

Significance: I think the significance of the paper is a little unclear -- see below

**Weaknesses:**

1.  It's not clear to me what is the motivation for finding these vulnerable regions.  What will we do with them?  If the goal is explainability, why is this method conceptually better than other alternatives?

2.  As is hinted at in the paper, there is a conceptual problem with the idea of using groups of one-pixel adversarial attacks as a kind of interpretability method: as images get higher and higher resolution, the influence of each individual pixel get smaller, so one-pixel attacks get harder to find.  Ideally, an interpretability method shouldn't fall apart as the resolution gets higher.

**Questions:**

How would the authors respond to the two weaknesses listed above?>

---

> ### Author Response · Authors · 2023-11-19
> **Response to Reviewer 4smo (part 1)**
>
> We greatly appreciate your insightful and constructive feedback. We address your concerns in the following discussion.
> ___
> >**Question1:** 'Traditional norm-based adversarial example generation algorithms, due to their lack of spatial constraints, often distribute adversarial perturbations throughout images, making it hard to identify these specific vulnerable regions.' If I understand correctly, the proposed method does not have spatial constraints that encourage the found pixels to be close together.
>
> **Answer:** You are right that our algorithm does not have spatial constraints that encourage the found pixels to be close together.  We provide a more detailed explanation in the context of the comparison with traditional norm-based adversarial example generation algorithms:
> - **Traditional Norm-Based Attacks:** In traditional norm-based adversarial attacks, each adversarial example typically contains a large number of perturbed pixels. Without spatial constraints, these techniques distribute perturbations throughout the entire image, leading to changes of the image's classification prediction due to the combined effect of all perturbed pixels. Consequently, it becomes challenging to pinpoint image regions that are specifically vulnerable to perturbations.  The primary challenge lies not in the distribution of perturbed pixels but in identifying the vulnerable regions and assessing their corresponding levels of vulnerability.
>
> - **Our Diverse One-Pixel Attack:**  Unlike traditional norm-based adversarial attack algorithms, our method identifies vulnerable regions using a variety of one-pixel adversarial examples. The vulnerable regions are formed by diversely located one-pixel perturbations.  The prediction change in each adversarial example can be directly attributed to the specific perturbed pixel. This feature of our algorithm permits the identification of vulnerable areas without relying on spatial constraints. It enhances our ability to not only detect specific susceptible regions but also accurately assess their vulnerability, setting the stage for in-depth further analysis, paving the way for a detailed subsequent analysis.
> ___

---

> ### Author Response · Authors · 2023-11-19
> **Response to Reviewer 4smo (part 2)**
>
> >**Question2: It's not clear to me what is the motivation for finding these vulnerable regions. What will we do with them? If the goal is explainability, why is this method conceptually better than other alternatives?**
>
> **Answer:** We genuinely appreciate the reviewer's insightful question regarding the motivation and potential applications of our work. For the motivation of our work:
> - **The motivation behind our work is to advance the field of explainable DNNs from another perspective - vulnerability by adopting a novel approach.** Traditional explainable DNN methods primarily concentrate on identifying areas in images that are most relevant to the model's predictions. In contrast, our research focuses on uncovering regions that are exceptionally sensitive to minor perturbations. This approach helps us uncover features within these regions that may be less reliable or robust. By validating and studying these vulnerable areas, we aim to deepen our understanding of the unexpected behaviors exhibited by DNNs. This shift in focus from simply interpreting model predictions to the underlying vulnerabilities represents the core motivation of our research.
>
> As mentioned above, our research primarily focuses on identifying vulnerable regions. For the question `what will we do with them', we give following explanation for the potential application.
> - **Enhance the Understanding of DNN's Vulnerability:** By examining which areas of an image, when altered, lead to misclassifications, we gain insights into the  features DNNs rely on for decision-making and their reliability. Understanding the conditions under which a DNN is prone to failure allows us to better grasp its limitations. This enhanced understanding is pivotal in addressing and mitigating vulnerabilities in DNNs.
>
> - **Enhance the Robustness of Deep Models:** Vulnerable regions within DNNs are areas in the input space highly sensitive to minor perturbations, which can significantly affect the network's predictions. Identifying and addressing these regions is crucial for enhancing the model’s defense against adversarial attacks. Employing certain strategies, in conjunction with an understanding of these vulnerable regions, may lead to a better way to enhance the robustness of deep models:
>     - Input Preprocessing: Applying preprocessing techniques to potentially vulnerable areas of images before their processing by the model can help mitigate the impact of possible perturbations in these high-risk regions.
>     - Ensemble Methods: We can integrate various models or defensive strategies into an ensemble. This ensemble will focus on addressing vulnerabilities across different regions, thereby improving its robustness against adversarial attacks.
>     - Adversarial Training: We can incorporate adversarial examples specifically designed to target identified vulnerable regions for improving the effectiveness of adversarial training, thereby enhancing the model's resilience.
>
>    These approaches of detecting and reinforcing model vulnerabilities can be utilized to enable DNNs to preserve their performance stability when faced with adversarial attacks.
>
> For the question `If the goal is explainability, why is this method conceptually better than other alternatives?', we have the following explanations:
> - Departing from traditional techniques ([1,2,3]), our method aims to enhance the comprehensibility of deep neural networks (DNNs) by focusing on vulnerability rather than relevance.  While conventional
> explainable DNNs concentrate on pinpointing image parts crucial for a model’s predictions, we aim to identify vulnerable regions where the features are not reliable.  This shift in focus provides a deeper, more fine-grained understanding of the model's weaknesses and limitations, which is crucial for developing more robust and reliable DNNs. This approach not only provides
> insight into where and when DNNs may fail, offering a superior understanding of model vulnerabilities compared to traditional methods.
>
> In summary, our research aims to enhance the comprehensibility of DNNs by focusing on vulnerability.   By identifying vulnerable regions and their potential applications, we aim to provide valuable insights into weakness and potential solutions of DNNs.
>
> [1] Matthew D Zeiler and Rob Fergus. Visualizing and understanding convolutional networks. In
> European Conference on Computer Vision, pp. 818–833. Springer, 2014
>
> [2] Vitali Petsiuk, Abir Das, and Kate Saenko. Rise: Randomized input sampling for explanation of
> black-box models. arXiv preprint arXiv:1806.07421, 2018
>
> [3] Ruth Fong, Mandela Patrick, and Andrea Vedaldi. Understanding deep networks via extremal per-
> turbations and smooth masks. In Proceedings of the IEEE/CVF International Conference on
> Computer Vision, pp. 2950–2958, 2019

---

> ### Author Response · Authors · 2023-11-19
> **Response to Reviewer 4smo (part 3)**
>
> >**Question3:** As is hinted at in the paper, there is a conceptual problem with the idea of using groups of one-pixel adversarial attacks as a kind of interpretability method: as images get higher and higher resolution, the influence of each individual pixel get smaller, so one-pixel attacks get harder to find. Ideally, an interpretability method shouldn't fall apart as the resolution gets higher.
>
> **Answer:** We are grateful for the reviewer's astute observation regarding high-resolution images.
> In addressing the concern that our interpretability method, which relies on groups of one-pixel adversarial attacks, might become less effective with increasing image resolution, we offer the following clarifications:
> - **Clarification why diverse *one-pixel* perturbations:**  Our method innovatively utilizes a variety of adversarial examples to enhance the interpretability of deep learning models, with a specific focus on meticulous, pixel-level analysis of vulnerabilities. Central to our strategy is the deliberate emphasis on scenarios involving one-pixel attacks. We concentrate on detecting vulnerable areas within images, critical for the recognition tasks of deep neural networks (DNNs). These areas are characterized by variably located single-pixel perturbations. However, we recognize that in the case of high-resolution images, the influence of each individual pixel is reduced, making such one-pixel attacks more complex to implement. To address these complexities inherent in high-resolution images, we have devised an adaptive strategies.
> - **Adaptation to the diverse patch attack  to overcome the limitation for higher resolution image:** Our algorithm is adaptable to higher-resolution images by incorporating a greater number of perturbed pixels in each adversarial example. This adaptation necessitates the use of spatial constraints to accurately represent vulnerabilities in specific areas, such as using small image patches. In the context of patch attacks, each candidate solution, can be modified to a tuple containing the coordinates of the top-left pixel of the patch and the RGB values of different perturbed pixels within the patch. As a result, the tuple contains $2+3 \times n$ elements, where $n$ represents the number of perturbed pixels.
>
>
>     This adapted approach has been validated on the ImageNet dataset with images resized to 224x224 pixels, using 4x4 pixel adversarial patches. (We have included the details of this experiment in **Appendix A.6** for further reference.) The results are presented in Table 1. This, however, shifts our analysis focus from individual pixels to patch-level analysis, balancing the trade-off between a higher success rate and the granularity of the vulnerability assessment.  Compared to the one-pixel attack scenario, there is no additional cost needed. The experiment results is presented in Table 1. For AlexNet, our proposed approach identified 95.2 distinct patches, collectively covering 1106.2 pixels. It is important to note that the size of the region does not equate to the product of the number of diverse patches and the size of each patch (e.g., $1106.2 \neq 95.2 \times 16$), which indicates overlapping among the patches. In the case of ResNet50, which exhibits a lower success rate compared to AlexNet, we found 174.8 different patches that cover a total of 1920.5 pixels.   These results demonstrated that our algorithm can be effectively adapted for high-resolution images with only minor adjustments. We leave a comprehensive analysis of these discovered vulnerable regions for our future work.
>
> **Table 1.** Comparison of vulnerable region discovery using diverse one pixels and diverse 4x4 pixel adversarial patches on ImageNet with our algorithm. Region size refers to the average number of different perturbed pixels covered in each successful attack.
> |                   | Perturbation Level | Success Rate | Region Size |
> |-------------------|--------------------|--------------|-------------|
> | AlexNet (ours)    | one pixel          | 11.7%        | 20.5        |
> | ResNet50 (ours)   | one pixel          | 10.7%        | 20.6        |
> | AlexNet (ours)    | 4x4 pixel patch    | 40%          | 1106.2      |
> | ResNet50 (ours)   | 4x4 pixel patch    | 28%          | 1920.5      |
>
>
> In summary, our algorithm can overcome resolution limitations with suitable adaptation. We leave a comprehensive analysis on higher resolution images for further work.

---

> ### Author Response · Authors · 2023-11-22
> **A kind reminder regarding our response**
>
> Dear Reviewer 4smo,
>
> We apologize for any inconvenience our request may cause during your busy schedule.   As the rebuttal phase is drawing to a close, we would be grateful if you could find a moment to review our responses.  We have made every effort to address the concerns you raised thoroughly and thoughtfully.
> If  you have any other questions, we hope you can communicate with us again. Your insights are crucial in helping us refine and improve our research.
>
> Looking forward to your reply.
>
> Thank you!
>
> Best regards,
> Authors of Paper ID3295

---

### Meta-Review · Area_Chair_4f5z · 2023-12-12

**Metareview:**

This paper introduces an evolutionary algorithm as an extension to Su et al. 2019 to spatially diverse sets of one-pixel attacks. The paper focuses most of the analysis on Cifar10, but also extends to pixel groups in larger images.

The strength of the paper is an extension of an existing algorithm to a slightly different use-case and further connecting model interpretability to adversarial examples.

On the weaknesses side, the novelty is limited, the motivation is lacking what exactly one would do with the identified regions and Cifar10 is too small of a dataset (in number of classes, image size, and network size applied to it) to be of interest to the community at this point. The motivation needs to be fleshed out more and a clear use-case needs to be presented where this framework enables a non-trivial advantage over one-pixel attacks or Lp-based attacks. I certainly think this can be done, but in its current form, the paper is not ready for publication at ICLR.

**Justification For Why Not Higher Score:**

The paper focuses too much on toy settings with little grounding in actual ML problems which the paper does claim to address by discussing interpretability and model vulnerability. Furthermore, much of the analysis focuses on Cifar10, which is a highly unrealistic setting. This paper requires a stronger motivation to be accepted and to demonstrate how it improves over Lp-bounded and one pixel attacks that have been around for many years.

**Justification For Why Not Lower Score:**

No lower score exists.

---

### Decision · Program_Chairs · 2024-01-16

Reject